# MERMAIDE: Learning to Align Learners using Model-Based Meta-Learning

## Abstract

Designing mechanisms like auctions or taxation policies can be formulated as a general-sum game between a principal and a self-interested *learning* agent. The principal aims to induce desirable outcomes in such games and may do so, for example, by dynamically intervening on the agent's learning objective. The intervention policy should generalize well to agents with unseen learning behaviors; in the real world, the principal may not know the agent's learning algorithm nor its rewards. Moreover, interventions may be costly, e.g., enforcing a tax might require extra labor; hence, interventions should be few-shot adaptable (only needs to retrain on few agents at test-time) and cost-efficient (uses few interventions). Here, we introduce a model-based meta-learning framework to train a principal that can quickly adapt when facing out-of-distribution agents with different learning strategies and reward functions. First, in a simple Stackelberg game between the principal and a greedy agent, we show that meta-learning allows adapting to the theoretically known and appropriate Stackelberg equilibrium at meta-test time, with few interactions with the agent. Second, we show empirically that our approach yields strong meta-test time performance against bandit agents with various unseen explore-exploit behaviors. Finally, we outperform baselines that separately use either meta-learning or agent behavior modeling to learn a cost-effective intervention policy that is $K$-shot adaptable with only partial agent information.

## 1 Introduction

General-sum games provide a framework to study diverse applications involving a principal that aims to incentivize an adaptive agent (both are learners) to achieve the principal's goal, e.g., maximizing revenue in auctions (Milgrom & Milgrom, 2004), optimizing social welfare with economic policy (Zheng et al., 2022), or optimizing skill acquisition in personalized education (Maghsudi et al., 2021). In this work, we focus on a principal that directly intervenes on the rewards of the agent. For instance, a government may want to incentivize the use of environmentally-friendly ("clean") products by levying green taxes, but needs to understand how people (strategically) change their consumption behavior as taxes change. Here, existing models of human adaptation that assume rational learning (or use simplified models of bounded rationality) often do not suffice. Hence, interacting with the agents is required to *learn* (how they change) their behavior, but such interactions are not "free". For example, a tax policy may require effort to apply it fairly and to measure its impact on consumers.

To mitigate the need for costly real-world interactions, we can use simulations with deep reinforcement learning (RL) agents. This is an attractive solution framework: deep neural network behavioral models are expressive enough to emulate real-world entities and simulations can be run safely and as often as needed. Moreover, we can use deep RL to learn intervention policies that are effective even in the face of complex agent behaviors in *sequential* general-sum games.

However, this approach also faces several challenges. When deploying the learned policies in the real world, interventions can typically only be applied a few times, due to implementation costs, and rarely under identical circumstances; in contrast to simulations, we cannot reset the real world. Even though principals may adapt their policies to new conditions, they cannot realistically know the true rewards or learning strategy of the agent. Hence, our goal is to learn policies in general-sum games

that 1) perform well even when agents learn, 2) can be quickly adapted, 3) are robust to distribution shifts in agent behaviors, and 4) are effective despite having only *partial information*.

**Contributions.** To address these challenges, we propose MERMAIDE (Meta-learning for Model-based Adaptive Incentive Design), a deep RL approach that 1) learns a world model and 2) uses gradient-based meta-learning to learn a principal policy that can be quickly adapted to perform well on unseen test agents. We consider two-player general-sum games between a *principal* and an *agent* wherein the principal intervenes *at a cost* on the agent's learning process to incentivize the agents to learn to act to achieve the principal's objective. We assume that the agent behaves in a first-order strategic manner and the principal in a second-order strategic manner. Here, the agents optimize their experienced rewards and minimize their regret, but do not account for their influence on the principal's actions. In contrast, the principal intervenes explicitly as to influence the agent's actions.

We first analyze the one-shot adaptation performance of a meta-learned principal in a matrix game setting, under both perfect and noisy observations for the agent and the principal. We show that meta-training reliably finds solutions that one-shot adapt well, and characterize how the principal's out-of-distribution performance depends on its observable information about the agent.

We next develop and empirically verify these insights with more adaptive agents and propose MERMAIDE which finds well-performing reward intervention policies in the sequential bandit setting. Here, MERMAIDE performs well against out-of-distribution bandit learners, with test-time performance and robustness depending on the agents' level of exploration and their pessimism in the face of uncertainty, confirming and extending the analysis and conclusions from the single-round setting.

## 2 RELATED WORK

**Bilevel optimization.** Learning a mechanism with agents who also learn is a bilevel optimization problem, which is NP-hard (Ben-Ayed & Blair, 1990; Sinha et al., 2017). Possible solution techniques include branch-and-bound and trust regions (Colson et al., 2007). In particular, solving bilevel optimization using joint learning of the mechanism and the agents can be unstable, as the agents continuously adapt their behavior to changes in the mechanism. This can be stabilized using curriculum learning (Zheng et al., 2020), but generally bilevel problems remain challenging, especially with nonlinear objectives or constraints.

**Meta-learning and distribution shift.** In recent years, gradient-based meta-learning has proven effective in learning initializations for complex policy models that generalize well to unseen tasks (Finn et al., 2017a; Nagabandi et al., 2018). Luketina et al. (2022) showed that context-conditioned meta-gradients are effective for adapting in environments with controlled sources of non-stationarity, but they do not account for non-stationarity from interactions between strategic agents that learn. Prior works in imitation learning (Argall et al., 2009) and inverse RL (Abbeel & Ng, 2004) assume access to expert demonstrations with a fixed policy that the (RL) agent wants to emulate. In contrast, our principal aims to learn a policy that can strategically alter the behavior of such demonstrators (our agents), who are themselves learning during an episode of the demonstration. Recently, Boutilier et al. (2020) studied meta-learning for bandit policies, while Guo et al. (2021) introduced the inverse bandit setup for learning from low-regret demonstrators. However, these works do not consider shifts in the bandit learning algorithm between training and test time.

**Modeling agents.** A key challenge in multi-agent learning is that each agent experiences a non-stationary environment if other agents are learning. As such, agents can benefit from having a *world model*, e.g., to know what the policy or value function of the other agents are. World models can stabilize multi-agent RL (Lowe et al., 2017) and enable higher-order learning methods (Foerster et al., 2018), and can be seen as a form of model-based RL. However, this may require a large amount of observational data or prior knowledge, which may be hard to acquire.

**Adaptive incentive design.** Principal-Agent problems (Eisenhardt, 1989) involve design of incentive structures, often under information asymmetry, but are usually not concerned with learning how to learn to incentivize across agents of different types. Pardoe et al. (2006) found that a form of meta-learning that adapts the learning process itself can design English auctions (sequential bidding)

that perform better with adaptive bidders who are loss-averse, and is still effective when the distribution of bidder behaviors (slowly) shifts. Our work expands on this theme by explicitly modeling agents that learn, considering shifts in the *learning algorithm* of the agents, and using deep RL with gradient-based meta-learning. The combination of these techniques enable learning incentivization policies that generalize well across more complex tasks.

## 3 LEARNING TO ALIGN AGENTS BY REWARD INCENTIVIZATION

**Overview.** We model a *principal* who aims to incentivize an *agent* to (learn to) execute the principal's preferred action. To do so, the principal can *intervene* and change the agent's rewards at a cost. Without interventions, the agent may learn to prefer an action different than the principal's.

For example, consider consumers who can use either environmentally "clean" or "dirty" goods. Indifferent at first, consumers may gradually learn to prefer dirty goods if those are consistently cheaper than clean ones, whereas the government may want them to prefer clean goods. Here, the agent's reward is the negative of the cost of consumption, for instance, and an intervention changes the price of goods through taxes or subsidies. If we have access to a simulation, the principal can compute an optimal intervention. However, the simulation might be inaccurate and real-world agents might behave differently. As an example of such test-time distribution shift, simulated agents may be fast to change their consumption preferences, while real agents may be slow. A "good" principal (trained in a simulation) could quickly be fine-tuned to intervene more in the latter case and adapt quickly if such behavior is observed during deployment.

In particular, we focus on *learning* a principal policy that needs to be adapted quickly following a single round of test-time game play (e.g., taxes and subsidies are deployed in the real world), and that is effective when the agent's learning algorithm differs from that seen during train-time.

We now formalize this setting. In this work, we focus on agents in a stateless environment for ease of exposition. For all variables and their meaning, see Tables 2 and 3 in the Appendix.

**The agent.** The agents are characterized by their action space $A$ and a base reward function $r : A \rightarrow \mathbb{R}$. We call it the base reward because the agent experiences an *intervened* reward

$$\tilde{r}_t(a_t) = r(a_t) + r'_t(a_t),\tag{1}$$

where the intervention $r'_t$ is provided externally (by the principal) for the agent action $a_t$. We index time as $t = 1, \ldots, T$. At each time step $t$, the agent's policy $\pi_t$ computes a distribution over its actions based on the observations for the agent up to timestep $t$ and executes $a_t \sim \pi_t$. We assume that the principal has a *preferred action* $a^*$ that the agent should execute, whereas the agent's optimal policy can prefer a different action than $a^*$ without intervention. Finally, at time $t$, the agent learns using an update rule $f : (\pi_t, a_t, \tilde{r}_t) \mapsto \pi_{t+1}$ to maximize the agent's intervened rewards, e.g., under UCB (Lai et al., 1985), $f$ updates the confidence bounds for the action selected at time $t$.

**The principal.** In this work, from the principal's point of view, the *world (environment)* consists of the agent who maximizes $\tilde{r}$. A standard assumption is that agents are rational and they may have a private state (referred to as its *type*) which the principal cannot see. Although the agent faces a stateless problem, *the principal faces a stateful problem with partial observability*. The full state $s \in S$ includes the principal's internal state $h_t^p$ (e.g., the principal's belief about the value of the private agent information), and all information about the agent, including its past actions, reward function, and policy model; often, the latter two are private.

More formally, the principal can be modeled as a POMDP $(S, o^p, A^p, r^p, \gamma, \mathcal{P})$. The observation function $o^p$ determines what part of a world state $s$ is visible to the principal, $A^p$ is its action space of interventions, $r^p$ is its reward, $\gamma$ is a discounting factor, and $\mathcal{P}$ are the environment dynamics, e.g., as caused by the agent's actions. At time $t$, the principal samples an action $\boldsymbol{a}_t^p \sim \pi^p\left(\boldsymbol{a}_t^p | o_{t-1}^p, h_{t-1}^p\right)$ which determines its intervention on each possible agent action $a$, i.e. $\boldsymbol{a}_t^p = \left[r'_1, \ldots, r'_{|A|}\right]$.

**Adaptive intervention policy learning** To model distribution shift at test time, we follow the meta-learning terminology (Finn et al., 2017b) and view each distinct agent as a *task* $\tau^i$. The principal has access to a *meta-train set of agents* $\tau^i \in \mathcal{T}_{\text{train}}; i = 1, \ldots, n_{\text{train}}$ and is evaluated on a *meta-test*

*set of agents $\tau^i \in \mathcal{T}_{\text{test}}; i = 1, \ldots, n_{\text{test}}$. We emphasize that during a task, both the principal and agent may learn and adapt, both at meta-train and meta-test time.*

Here, we focus on two key challenges: $K$-shot adaptation and distribution shift. First, the principal gets only $K$ episodes for fine-tuning for each meta-test task (but can train indefinitely for each meta-train task). Second, the principal faces two types of distribution shift: 1) across tasks and 2) intra-task non-stationarity. The meta-train and meta-test tasks may differ (significantly) in their temporal distribution of actions, e.g., due to different agent updates $f$ or the agent rewards $r_t$ being centered around different values (e.g., average price levels are higher in the real world vs in the simulation). Within a task, the agent's learning is affected by the principal's interventions that change its reward $\tilde{r}$. This gives rise to non-stationarity in the agent's environment, as its learning objective may shift over time. These forms of distribution shift distinguish our adaptive intervention policy learning setting from most prior work in meta-learning, which often assume stationarity within a task and also assume similar task distributions at meta-train and meta-test times.

**Objectives.** The principal's objective is to maximize how often meta-test-time agents choose $a^*$ during learning and have them converge to a policy that always chooses $a^*$. To do so, the principal aims to maximize the cost-adjusted test-time return $J^p_{\text{test}}\left(\pi^p, \pi^i\right) = \mathbb{E}_{\mathcal{T}_{\text{test}}}\left[\sum_{t=1}^{T} \gamma^{t-1}(r_t^p - \alpha c_t)\right]$, where the agent executes its (optimal) policy $\pi^i\left[\pi^p\right]$ in response to $\pi^p$:

$$\arg\max_{\pi^p} \mathbb{E}_{\tau^i \in \mathcal{T}_{\text{test}}} \mathbb{E}_{\pi^p} \mathbb{E}_{\pi^i[\pi^p]}\left[\sum_{t=1}^{T} \gamma^{t-1}(r_t^p - \alpha c_t)\right], \quad r_t^p = \mathbf{1}\left[a_t = a^*\right], \quad \alpha > 0, \quad (2)$$

where the principal incurs a cost $c_t$ if it intervenes. A simple cost function is $c_t = \mathbf{1}\left[r'_t \neq 0\right]$, i.e., the cost is constant across non-trivial interventions, where $\alpha > 0$ is a constant. Note that if intervention were free ($c_t = 0$), a trivial solution is to always add a large $r'\left(a^*\right) \gg 0$ for its preferred action $a^*$, such that it always yields the highest reward. Hence, we focus on learning non-trivial strategies when intervention is costly, which forces the principal to strategically alter the agent's learning behavior.

During an episode of $T$ time steps, each agent $i$ starts with a uniformly initialized action probability distribution $\pi_0^i$ and optimizes $\pi_t^i$ subject to interventions $\pi^p$ to maximize its return: $\mathbb{E}_{\pi^i} \mathbb{E}_{\pi^p}\left[\sum_{t=1}^{T} \tilde{r}_t^i\left(a_t^i, a_t^p\right)\right]$. Here, we assume that $T$ and $\gamma$ are sufficiently large so the agent converges to its optimal policy under $\tilde{r}$, using its learning algorithm $f$. That is, we assume that the objective in Eq. (2) is sufficient to describe the principal's objective of ensuring the agent converges to preferring $a^*$ at some $t < T$.

In the $K$-shot adaptation setting, at meta-test time, the principal gets $K$ episodes to interact with any agent, each episode of length $T$ steps. The principal has a fixed policy during an episode and it can update its policy at the end of an episode. The agent is reset across episodes, and within each episode, the agent follows its own learning strategy in response to the principal's interventions. On the $K + 1^{\text{th}}$ episode, the principal evaluates its $K$-shot adapted policy on the agent. Note this assumes that the principal has a separate copy of the meta-test time agent for evaluation.

## 4 ANALYSIS IN THE MATRIX GAME SETTING

We first study robust adaptive intervention policy learning with strategic agents using a simple 2-player game between a principal and an agent. The agent's actions are "cooperate" and "defect", while the principal can choose whether or not to intervene. Assuming the row player is the agent and the column player is the principal, the $2 \times 2$ payoff matrix is given by

$$\begin{array}{cc} & \begin{array}{cc} \text{No intervention} & \text{Intervene} \end{array} \\ \begin{array}{c} \text{Cooperate} \\ \text{Defect} \end{array} & \left(\begin{array}{cc} u, 1 & u+1, 1-c \\ 1-u, 0 & -u, -c \end{array}\right), \end{array} \quad (3)$$

where $u \in (0, 1)$ and $c$ is the cost of intervention ($c < 1$). The principal prefers cooperation: it gets 1 if the agent cooperates and 0 if the agent defects (minus the cost $c$ if it intervenes). The agent's base payoff $u$ is its type. Notice that an intervention incentivizes the agent to cooperate ($u + 1 > -u$).

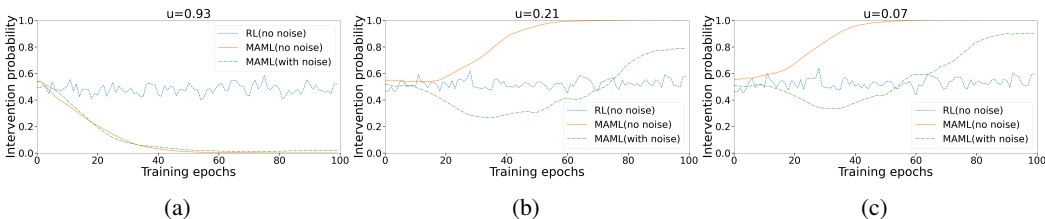

(a)                              (b)                              (c)

Figure 1: **Single round game.** REINFORCE (RL) does not adapt to expected Stackelberg equilibrium during evaluation. MAML's adaptability suffers under observation noise.

It is natural to consider the Stackelberg setting where the principal is the leader and acts first (intervene or not), and the agent acts second (Von Stackelberg, 2010). Our goal is to learn an intervention policy that can adapt to different agent types and find the Stackelberg equilibrium. We now analyze three scenarios with increasing difficulty:

1. First, we assume that the principal knows $u$. Here, there is a unique Stackelberg equilibrium at (Cooperate, No Intervention) when $u \geq \frac{1}{2}$, and at (Cooperate, Intervene) when $u < \frac{1}{2}$.

2. Second, the principal can observe a noisy version of $u$. In both these cases, the agent first observes the principal's action and plays the best response according to its payoff matrix.

3. Finally, we consider a repeated multi-stage game where the agent cannot observe the principal's action. Instead, we assume that the agent keeps a running average for the experienced payoffs for each of its actions. In a single-round setting this would correspond to the principal committing to a mixed action and then the agent choosing its best response. When $u \geq \frac{1}{2}$, the Stackelberg equilibrium occurs at (Cooperate, No Intervention). When $u < \frac{1}{2}$, at the Stackelberg equilibrium for this game the principal has a mixed action where it chooses to intervene for $\frac{2u+1}{2}$ fraction of times and the agent chooses to always cooperate.

Given this equilibrium analysis, we learn a neural network policy for the principal that predicts its probability of intervention and compare the behavior of the learned policy when trained using standard policy gradients (RL) versus meta-learning (MAML, (Finn et al., 2017b)). We set $c = 0.75$.

**With perfect observability.** Here, we study whether meta-learning finds a better initialization $\theta_{\text{meta}}$ for adaptation on unseen agents. In this setting, we assume that the principal observes an agent's exact payoff parameter $u$. It learns a stochastic policy $\pi_\theta^p(u)$ which determines its probability of intervening in a single-round game with an agent of type $u$. Given a set of training agents with different types $u \sim \mathcal{U}(0,1)$, for each $u$, the principal learns the optimal policy parameters $\theta^*(u) = \arg\max_\theta \mathbb{E}_{a^p \sim \pi_\theta^p(u)}[r^p(a^p)]$, where $r^p(a^p)$ is the principal's payoff for action $a^p$ with agent $u$. The planner then learns $\theta_{\text{meta}}$ using the meta-learning algorithm in Appendix B and one-shot adapts on a meta-test set of agents with different $u$s than at training. Note that we're studying the quality of the initialization, not the generalization performance of an already trained policy. Fig. 1 shows the principal's meta-test time probability of intervening with 3 different agents from the test set, across training epochs. The principal and agent should be at different Stackelberg equilibria depending on the type $u$, as discussed above. We see that a principal trained from scratch on the test agents using standard policy gradients is unable to adapt to different agents in a single-shot adaptation setting. In contrast, with meta-learning, the principal learns a better policy that is one-shot adaptable to agents of different types and converges to the correct Stackelberg equilibrium at meta-test time.

**With noisy observations for the principal.** Here, we emulate a principal with partial observability of the agent, by letting the principal observe $u$ with added i.i.d. Gaussian noise. The agent can see all payoffs and chooses the best response to achieve a Stackelberg equilibrium. Fig. 1 shows that with noisy observations, the meta-learned principal policy requires more training time to be one-shot adaptable to the optimal intervention policy. This empirically indicates the increased difficulty of learning an adaptive intervention policy due to incomplete information about the agent, especially under limited adaptation time with unseen agents. It therefore motivates us to adopt a *model-based* approach for the principal to better estimate the agent type and learn an adaptive intervention policy.

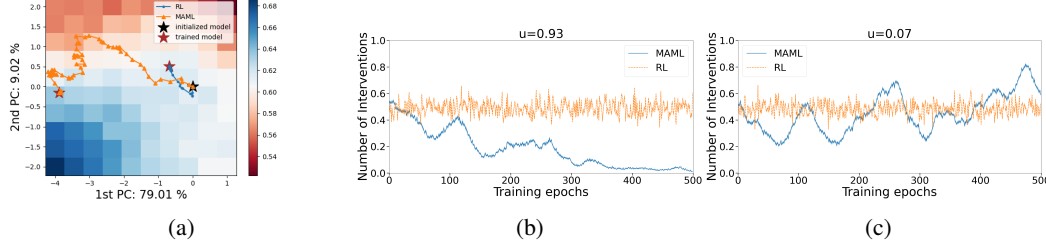

Figure 2: **Multi-round game.** (a) Principal's optimization trajectory in the expected payoff landscape during training. Axes are PCA directions in the policy parameter space. (b)(c) MAML adapts (single shot) to Stackelberg equilibrium with a best response agent in a simplified form of MD.

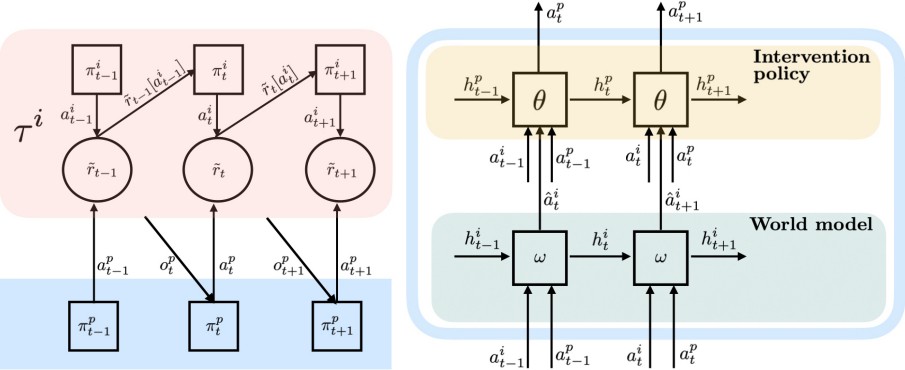

Figure 3: Overview of **MERMAIDE.** Left: Flow of principal and agent observables, rewards, and actions. Right: The principal's world model and intervention policy. Also see Algorithm 1.

Comparing Fig. 1b and Fig. 1c, we also see that when $u < \frac{1}{2}$, the difference in unintervened payoffs between the principal's preferred action $(u)$ and the agent's preferred action $(1 - u)$ also impacts the one-shot adaptability of the principal receiving noisy observations. This observation informs our analysis of the bandit setting in Section 6.

**Multi-round repeated game with noisy rewards.** In this setting, the principal and agent repeatedly play an iterated game over $T = 100$ steps. In each round, the principal observes the agent's type $u$ with added i.i.d. Gaussian noise. The agent cannot observe the principal's actions, and plays a best response for its current estimate of the action payoffs. Whenever the agent selects an action, it receives a noisy observation of the true payoff and updates its estimate. Compared to the single-round setting, here the agent's best response behavior may change across rounds in the game depending on its observed payoffs, giving rise to non-stationarity in the principal's environment. The planner, in turn, has to learn to intervene so that the agent's best response is to cooperate.

Fig. 2a compares the optimization trajectory followed using 1) standard policy gradients and 2) meta-learning for the principal. Starting from the same initialization, the meta-learned policy's parameters lie in a region of the payoff landscape with a higher expected value over the training agents. Moreover, Fig. 2b and Fig. 2c show the one-shot adaptability of the principal's policy for two different agent types at meta-test time. Meta-learning helps learn a better intervention strategy that is robust to the principal's observation noise as well as the agent's evolving best response strategy.

## 5 MERMAIDE: Learning to Align Learners

Motivated by our findings from Section 4, we now present MERMAIDE (Fig. 3), consisting of:

---

**Algorithm 1** MERMAIDE (Notations also in Table 3)

---

1: Initialize principal $(\theta_0, \omega_0)$, and hidden states $h_0^i, h_0^p$.
2: **for** meta-train epoch $e = 1, \ldots, E_{\text{train}}$ **do**
3:      Update world model parameters $\omega = \omega_e$ using Eq. (4).
4:      **for** agents (tasks) $i = 1, \ldots, n_{\text{train}}$ **do**
5:          Initialize agent: $(\mu^i, \pi_0^i)$, task specific principal policy parameter $\theta\left(\tau_0^i\right) = \theta_e$.
6:          **for** $k = 1, \ldots, K_{\text{train}}$ **do**            $\triangleright$ Inner loop for $K_{\text{train}}$ episodes.
7:              **for** time t $= 1, \ldots, T$ **do**     $\triangleright$ For each episode with $T$ principal-agent interactions
8:                  Predict $\hat{a}_t^i = \arg\max_{a_t^i} \hat{\pi}_\omega \left(a_t^i | a_{t-1}^i, a_{t-1}^p, h_{t-1}^i\right)$ using the world model.
9:                  Intervention: $\tilde{\mu}^i = \mu^i + a_t^p, \quad a_t^p \sim \pi_{\theta\left(\tau_k^i\right)}^p \left(a_t^p | a_{t-1}^i, a_{t-1}^p, \hat{a}_t^i, h_{t-1}^p\right)$.
10:                  Agent acts: $a_t^i \sim \pi_t^i$ and receives reward $r_t^i \sim \mathcal{N}\left(\tilde{\mu}^i, \sigma^2\right)$. $\pi_t^i \mapsto \pi_{t+1}^i$.
11:              Locally update $\theta\left(\tau_k^i\right) \mapsto \theta\left(\tau_{k+1}^i\right)$.            $\triangleright$ Using REINFORCE.
12:          **for** $t = 1, \ldots, T$ **do**            $\triangleright$ Rollout for meta-update; $\mathcal{D}_{\text{meta}}\left(\tau^i\right) = \{\}$
13:              Predict $\hat{a}_t^i = \arg\max_{a_t^i} \hat{\pi}_\omega \left(a_t^i | a_{t-1}^i, a_{t-1}^p, h_{t-1}^i\right)$ using the world model.
14:              Intervention: $\tilde{\mu}^i = \mu^i + a_t^p, \quad a_t^p \sim \pi_{\theta\left(\tau_{K_{\text{train}}}^i\right)}^p \left(a_t^p | a_{t-1}^i, a_{t-1}^p, \hat{a}_t^i, h_{t-1}^p\right)$.
15:              Agent acts: $a_t^i \sim \pi_t^i$, receives reward $r_t^i \sim \mathcal{N}\left(\tilde{\mu}^i, \sigma^2\right)$. Updates $\pi_t^i \mapsto \pi_{t+1}^i$.
16:              Collect $\mathcal{D}_{\text{meta}}\left(\tau^i\right) \cup \left\{a_t^i, a_t^p, \pi_{\theta\left(\tau_{K_{\text{train}}}^i\right)}^p\right\}$
17:      Meta-update $\theta_e \mapsto \theta_{e+1}$ using $\mathcal{D}_{\text{meta}} = \cup_{\tau^i} \mathcal{D}_{\text{meta}}\left(\tau^i\right)$.            $\triangleright$ Using MAML.

---

- a recurrent *world model* parameterized by $\omega$ that outputs a distribution over an agent $i$'s actions at the next time step $t$: $\hat{\pi}_\omega \left(a_t^i | a_{t-1}^i, a_{t-1}^p, h_{t-1}^i\right)$, conditioned on the planner's intervention and the observed agent action at $t-1$. $h_{t-1}^i$ is the hidden world model state.

- a recurrent *intervention policy* which outputs a distribution over interventions $a_t^p \sim \pi_\theta^p \left(a_t^p | a_{t-1}^i, a_{t-1}^p, \hat{a}_t^i, h_{t-1}^p\right)$, conditioned on its previous intervention, the observed agent action and the world model's predicted next agent action $\hat{a}_t^i = \max_a \hat{\pi}_\omega \left(a | a_{t-1}^i, a_{t-1}^p, h_{t-1}^i\right)$. $h_{t-1}^p$ is the hidden state of the policy network.

We train this using gradient-based meta-learning and RL, see Algorithm 1. Here, the principal maximizes the meta-train objective $J_{\text{train}}^p$ similar to the objective in Eq. (2). The base RL algorithm is REINFORCE (Williams, 1992) and the meta-learning update uses MAML (Finn et al., 2017b). The agent optimizes its cumulative intervened reward, see Section 6 for details. The world model $\hat{\pi}_\omega$ trains by maximizing the log-likelihood of the observed $a_t^i$, using Adam (Kingma & Ba, 2014):

$$\arg\max_\omega \mathbb{E}_{a^p \sim \pi^p} \mathbb{E}_{a^i \sim \pi^i} \left[\sum_{t=1}^T \log \hat{\pi}_\omega \left(a_t^i | a_{t-1}^i, a_{t-1}^p, h_{t-1}^i\right)\right]. \tag{4}$$

Note that the principal's parameters $\theta$ are updated after each $T$-step episode, while the agent continuously learns during each episode. Also, the agent is reset in between episodes. At time 0, the world model makes a prediction based on zero initialization. We use a single world model for all agents. At meta-test time, only the intervention policy is updated by one-shot adaptation to a new agent.

## 6 EXPERIMENTAL VALIDATION IN THE BANDIT SETTING

We now study a sequential general-sum game between the principal and an adaptive no-regret learner agent, modeled by an $|A|$-armed bandit instance with action set $A$ having base reward $\boldsymbol{r} = \left[r_1, \ldots, r_{|A|}\right]$. At each time step $t$, the agent chooses an arm $a$ and gets a reward sampled from $\mathcal{N}\left(r_a, \sigma^2\right)$. We assume $r_a \in (0, 1) \;\forall a$. The agent aims to maximize its cumulative reward over a horizon of $T$ steps. The agent can only observe the reward for the chosen action, and hence faces a explore-exploit dilemma addressed by bandit algorithms like UCB (Lai et al., 1985). We assume there is a unique arm $\tilde{a}$ with the highest base reward: $\tilde{a} = \arg\max_a r_a$, i.e., the agent's preferred action without any intervention.

**Costly interventions.** To analyze the effect of the cost of intervention $c_t$ on the principal's learnt policy, we assume that the principal decides among three different intervention levels $|r'| \in \{0, 0.5, 1\}$ such that $c_t = |r'|$. Across different bandit agent tasks $\tau^i$ with distinct base rewards $\boldsymbol{r}^i$ and reward gaps $\delta = \max_{a \in A} \boldsymbol{r}^i[a] - \boldsymbol{r}^i[a^*]$, the principal should learn to appropriately incentivize the agent while minimizing the total cost of intervening. We then define the experienced reward as:

$$\tilde{\boldsymbol{r}}_t[a^*] = \boldsymbol{r}^i[a^*] + r'_t; \quad \tilde{\boldsymbol{r}}_t[a] = \boldsymbol{r}^i[a] - r'_t, \quad \forall a \neq a^*, (a, a^* \in A). \tag{5}$$

Note that this ensures the agent always experiences an intervention, no matter which action it chooses. During each episode, the agent learns but the principal's policy is fixed; the principal can update its policy only at the end of each episode (Algorithm 1). Also, we assume that the principal can only observe the agent's actions $a_t^i$ but not its base reward $\boldsymbol{r}^i$ or policy update function $f^i$. We measure the performance of the principal using Eq. (2), with $\gamma = 1$.

**World model.** The world model predicts the agent's next *action* (given the principal's prior observations) to characterize the agent's behavior. We do not train the principal's world model to estimate the base rewards, because bandit agents with distinct base rewards could still execute the same sequence of actions, depending on the agent's explore-exploit algorithm and its observations.

**Challenges in the sequential setting.** Compared to the simple game setting in Section 4, principal's intervention policy learning with sequential (bandit) learners creates additional challenges:

- Bandit agents may follow different strategies for action selection to maximize their experienced reward. The agent's rate of exploration may be constant (e.g., $\epsilon$-greedy) or it can reduce with time (e.g., UCB) within an episode, depending on its observations. This creates a highly non-stationary environment for the principal wherein its decision to intervene must adapt to different explore-exploit behaviors for the same agent within an episode. When the agent explores a larger action space, it further exacerbates the challenges in estimating the agent's behavior since the principal only has partial information about the agent.

- Bandit agents are sequential learners and feedback $(a_t^i, \tilde{r}_t^i)$ can update the policy $\pi^i$ differently at different steps $t$. The update may depend on how optimistic (e.g., UCB) or pessimistic (e.g., EXP3) the bandit agents are about their reward estimates. Hence, an intervention $a^p$ may not equally incentivize the agent at different $t$. Since the principal's interventions have different costs, a strategic principal must decide *when* to intervene and *how much* ($|r'|$) depending on its observations of the agent's actions.

**Results.** In the following experiments, we use 15 bandit agents for training and 10 bandit agents for testing, each with different base rewards (both within and across train and test sets). $|A| = 10$. We consider two agent learning algorithms (UCB and $\epsilon$-greedy). In each experiment, the train and test agents use the same algorithm, but with different tendencies for exploration vs exploitation, determined by their exploration coefficients: $\beta \in \{0.17, 0.27, 0.42, 0.5, 0.67\}$ for UCB (higher $\beta$ gives more exploration) and $\epsilon \in \{0.1, 0.2, 0.3, 0.4, 0.5\}$ for $\epsilon$-greedy (higher $\epsilon$ gives more exploration). These constants were chosen such that they afford, on the average, the same number of exploratory actions when following either UCB or $\epsilon$-greedy strategy without any intervention (see Appendix B).

In Table 1, we show the one-shot adapted principal's score on each test set over $T = 200$ time steps. We compare MERMAIDE against 1) model-free baselines (MF-RL using REINFORCE and MF-MAML using MAML), as well as 2) REINFORCE with world model (WM-RL) (see Appendix B for details). We also include a "No Intervention" baseline to show how agents behave by default.

**Out-of-distribution performance.** Table 1 shows the principal's score when evaluated on test agents having a *different exploration constant* than train agents. Using meta-learning for the intervention policy (MF-MAML) and using a world model to predict the agent's behavior (WM-RL) both have advantages for training a robust and one-shot adaptable intervention policy. A world model is advantageous when 1) the test agent is more exploratory than the train set (e.g., $\epsilon = 0.1$ at training, $\epsilon = 0.4$ at test), or 2) the agent explores throughout an episode and is likely to often select actions other than the one with its current maximum mean reward estimate (e.g., $\epsilon = 0.5$ at training). Because we evaluate on $K = 1$, fine-tuning on only a single test-time episode, a trained world model provides a useful prior belief representation for the principal. Indeed, the MF-RL results show the

| Train on UCB, $\beta = 0.17$ | Test on $\beta = 0.17$ | $\beta = 0.27$ | $\beta = 0.42$ | $\beta = 0.5$ | $\beta = 0.67$ |
|---|---|---|---|---|---|
| *No intervention* | 3 (0) | 5 (0) | 8 (0) | 10 (0) | 12 (0) |
| MF-RL | 119 (2) | 109 (2) | 98 (2) | 90 (2) | 77 (1) |
| MF-MAML | 133 (2) | 125 (3) | 107 (1) | 97 (1) | 77 (0) |
| WM-RL | 123 (7) | 112 (6) | 100 (4) | 92 (2) | 75 (1) |
| MERMAIDE (ours) | **154 (2)** | **141 (1)** | **115 (1)** | **103 (0)** | **80 (1)** |
| **Train on $\epsilon$-greedy, $\epsilon = 0.1$** | $\epsilon = 0.1$ | $\epsilon = 0.2$ | $\epsilon = 0.3$ | $\epsilon = 0.4$ | $\epsilon = 0.5$ |
| *No intervention* | 3 (0) | 4 (1) | 7 (0) | 9 (1) | 11 (0) |
| MF-RL | 115 (5) | 94 (4) | 54 (19) | 39 (6) | 22 (9) |
| MF-MAML | 122 (4) | 97 (3) | 58 (5) | 40 (2) | 12 (1) |
| WM-RL | 115 (4) | 94 (5) | 70 (1) | 55 (3) | **38 (1)** |
| MERMAIDE (ours) | **134 (1)** | **108 (1)** | **85 (1)** | **57 (7)** | 29 (1) |
| **Train on UCB, $\beta = 0.67$** | $\beta = 0.17$ | $\beta = 0.27$ | $\beta = 0.42$ | $\beta = 0.5$ | $\beta = 0.67$ |
| *No intervention* | 3 (0) | 5 (0) | 8 (0) | 10 (0) | 12 (0) |
| MF-RL | 103 (3) | 101 (3) | 92 (2) | 85 (1) | 74 (1) |
| MF-MAML | 124 (2) | 116 (1) | 102 (1) | 94 (1) | 80 (1) |
| WM-RL | 100 (4) | 89 (0) | 85 (1) | 85 (1) | 74 (0) |
| MERMAIDE (ours) | **131 (2)** | **125 (2)** | **109 (1)** | **101 (1)** | **85 (1)** |
| **Train on $\epsilon$-greedy, $\epsilon = 0.5$** | $\epsilon = 0.1$ | $\epsilon = 0.2$ | $\epsilon = 0.3$ | $\epsilon = 0.4$ | $\epsilon = 0.5$ |
| *No intervention* | 3 (0) | 4 (1) | 7 (0) | 9 (1) | 11 (0) |
| MF-RL | 4 (5) | 2 (3) | 5 (0) | 11 (5) | 7 (1) |
| MF-MAML | 2 (0) | 4 (0) | 6 (0) | 8 (1) | 11 (1) |
| WM-RL | 102 (6) | 79 (10) | 68 (3) | 47 (1) | 30 (2) |
| MERMAIDE (ours) | 87 (42) | **102 (3)** | **78 (6)** | **69 (1)** | **46 (2)** |

Table 1: **Test-time principal mean and standard error scores across 3 random seeds.** Left column: Principal's algorithm (e.g., MERMAIDE), training agent type (e.g., UCB with $\beta = 0.17$). Other columns: Test-time scores on agents with the same algorithm, but different hyperparameters.

hidden state representation of the model-free principal might be unable to adapt to high environment non-stationarity without a trained next-agent-action world model.

Compared to an $\epsilon$-greedy agent, the UCB agent explores mostly at the start of an episode, for all $\beta$. Hence, with UCB agents, the principal learns an effective one-shot adaptable intervention policy using meta-learning (MF-MAML) only (even without a world model), as the agents cause less distribution shift across different $c$. It further emphasizes the effectiveness of meta-learning for adaptive policy learning: unlike MF-MAML, neither the world model nor the intervention policy is meta-learned in WM-RL. Moreover, it also shows that for the same amount of distribution shift (characterized in Appendix B), the relative benefit of a world model or meta-learning the principal's policy depends on the nature of the agent's exploration strategy (which is unknown to the principal).

In all, these results show that MERMAIDE combines the best of both techniques: the principal obtains a higher score across agents with different learning algorithms and explore-exploit behaviors.

**Agent exploration vs intervention cost.** In order to intervene effectively, the principal should learn *when* to intervene and *how much* to incentivize the agent while minimizing its incurred cost. This is a challenging learning problem for the principal not just during meta-training, but more so during one-shot adaptation at meta-test time. Bandit algorithms like EXP3 (Auer et al., 2002) use pessimism in the face of uncertainty, and encourage continued exploration. This increases the non-stationarity for the principal. In order to effectively incentivize such agents to prefer $a^*$, the principal needs to accurately predict the agent's policy from its observations; otherwise it can incur a high cost for intervening ineffectively *and* lowering its score, and learn to stop intervening. Indeed, our results when training on $\epsilon = 0.5$-greedy agents show that the MF-RL and MF-MAML principal stop intervening. In contrast, in that setting, MERMAIDE learns an effective intervention policy that outperforms all baselines, even under distribution shift between meta-train and meta-test agents.

## 7 DISCUSSION AND FUTURE WORK

We have shown that MERMAIDE is an effective framework to learn principal intervention policies that generalize well to agents with unseen learning behavior. Future work could extend MERMAIDE to settings with multiple learning agents who may coordinate, compete, or a combination thereof. Moreover, it is interesting to extend MERMAIDE to agents that adapt adversarially to the principal's intervention policy, which poses a challenging non-stationary problem for the principal.

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

| Variable | Symbol |
|---|---|
| Time | $t$ |
| Principal | $p$ |
| Agent | $i$ |
| State | $s$ |
| State vector | $\boldsymbol{s}$ |
| State space | $S$ |
| Agent's action space | $A$ |
| Principal's action space | $A^p$ |
| Action sequence | $a_{1:T} = \{a_1, a_2, \ldots, a_T\}$ |
| Agent $i$'s reward sequence | $\tilde{r}^i_{1:T} = \{\tilde{r}^i_1, \ldots, \tilde{r}^i_T\}$ |
| Principal's reward sequence | $r^p_{1:T} = \{r^p_1, \ldots, r^p_T\}$ |
| Transition function | $\mathcal{P}$ |
| Agent $i$'s policy | $\pi^i$ |
| Principal's intervention policy | $\pi^p$ |
| Agent's mean estimate of intervened rewards for action $a$ | $\tilde{\mu}_a$ |
| Number of adaptation steps | $K$ |
| Number of meta-tasks for the planner | $N$ |
| Principal's history of interventions and observed agent actions upto time $t$ | $\mathcal{H}^p_t = \left\{a^p_1, a^i_1, a^p_2, a^i_2, \ldots, a^p_{t-1}, a^i_{t-1}\right\}$ |
| Agent's history of actions taken and rewards observed upto time $t$ | $\mathcal{H}^i_t = \left\{a^i_1, \tilde{r}^i_1, a^i_2, \tilde{r}^i_2, \ldots, a^i_{t-1}, \tilde{r}^i_{t-1}\right\}$ |

Table 2: **Overview of notation.**

| | |
|---|---|
| Principal's policy parameter | $\theta \in \Theta$ |
| Agent $i$'s learning algorithm | $f^i \in \mathcal{F}$ |
| Agent $i$'s true action mean rewards | $\mu^i \sim \mathcal{U}$ |
| Agent $i$'s intervened action mean rewards | $\tilde{\mu}^i$ |
| Principal's action at time $t$ | $a^p_t \sim \pi^p_\theta \left(a^p_t | a^i_{t-1}, a^p_{t-1}, \hat{a}^i_t, h^p_{t-1}\right)$ |
| Hidden state space of the principal's recurrent world model | $H$ |
| Agent's action at time $t$ | $a^i_t \sim \pi^i_t \left(a^i_t | \mathcal{H}^i_t\right), t = 1, \ldots, T$ |
| Agent's reward at time $t$ | $r^i_t \sim \mathcal{N}\left(\tilde{\mu}^i, \sigma^2\right)$ |
| Principal's world model estimate of the agent's action probability distribution | $\hat{\pi}^i_\omega : A \times A^p \times H \to \Delta\left(A\right),\ \hat{\pi}^i_{\omega,0} \in A$ |
| Principal's world model estimate of the latent state of the environment | $g^i_\omega : A \times A^p \times H \to H,\ g^i_{\omega,0} \in \Delta\left(H\right)$ |
| Principal's world model hidden state embeding in the LSTM architecture | $h^i_t = g^i_\omega \left(a^i_{t-1}, a^p_{t-1}, h_{t-1}\right),\ t = 2, \ldots, T\ h^i_1 = g^i_{\omega,0}$ |

Table 3: **Notation for MERMAIDE** See Section Section 5 for their use.

## A  NOTATION

For an overview of all symbols and variables used in this work, see Table 2 and Table 3.

# B  ADDITIONAL RESULTS

## B.1  DESCRIPTION OF THE BANDIT ALGORITHMS

We provide a brief overview of the learning algorithms referred to in Section 6.

**UCB.**  This is an Upper Confidence Bound based exploration-exploitation algorithm that follows the principle of optimism in the face of uncertainty. At each time step $t$, the bandit agent selects an action

$$a_t = \arg\max_a \tilde{\mu}_a + \beta \sqrt{\frac{\log t}{n_a}} \tag{6}$$

where $n_a$ is the number of steps until $t$ in which it previously selected the action $a$, $\tilde{\mu}_a$ is its corresponding mean estimate for the experienced rewards $\tilde{r}$ for action $a$ and $\beta$ is the exploration constant that balances the amount of exploration vs. exploitation across a time horizon $T$. A higher value of $\beta$ makes the agent less optimistic and explore its action space more. The UCB agent's tendency to explore is also affected by the difference in the mean reward estimates of its actions. In the context of our mechanism design problem formulation, if the UCB agent has a larger value of $\delta = \max_a r_a - r_{a^*}$, without any intervention at the beginning of an episode, its confidence bounds would quickly converge to exploiting the action $\arg\max_a r_a$. So a principal that intervenes only towards the later stages of an episode with this agent would have to provide much more incentives (higher $r'$) to alter the agent's preferred action to be $a^*$, thus incurring a larger cost $c$ as compared to a principal that intervenes more at the beginning of an episode when the UCB agent is still exploring its action space. This is also illustrated in Section 4 with a simpler best response agent in the single round game setting. As shown in Fig. 1b and Fig. 1c, under observation noise (partial information), the meta-trained principal has a better one-shot meta-test-time performance when the agent's base payoff has a higher difference between the principal's preferred action and the agent's intrinsic preference without any intervention.

**$\epsilon$-greedy.**  A simple exploration-exploitation strategy in the bandit setting is the $\epsilon$-greedy rule (Sutton & Barto, 1998) wherein the agent selects with probability $1 - \epsilon$ the action $a_t = \arg\max_a \tilde{\mu}_a$ and with probability $\epsilon$ it selects a random action. In our setting, we consider $\epsilon$ to be constant during an episode, which results in a uniform exploration rate throughout. In contrast to the UCB agent, the $\epsilon$-greedy algorithm simulates a less optimistic, more exploratory agent for which the principal requires a robust belief representation of the agent's predicted behavior conditioned on the principal's past observations (Table 1). Since there is a uniform exploration rate for the agent, the principal has to continue intervening intermittently throughout an episode, especially when $\delta$ is large and the agent could obtain a higher reward for an action $a \neq a^*$ by exploring its action space when the principal does not intervene.

**EXP3.**  The Exponential-weight algorithm for Exploration and Exploitation (EXP3) (Auer et al., 2002) follows a more pessimistic approach to exploration-exploitation in the bandit setting. It maintains a set of weights for each agent action $a \in A$ which are updated using the experienced rewards $\tilde{r}$ as follows:

$$\pi_t(a_t) = \frac{w}{|A|} + (1 - w) \frac{\eta \exp\left(S_{a_t,t}\right)}{\sum_{a_t \in |A|} \eta \exp\left(S_{a_t,t}\right)}, \tag{7}$$

where

$$S_{a_t,t} = \sum_{l=1}^{t} \mathbf{1}\left\{a_l = a_t\right\} \frac{\tilde{r}_{a_t,l}}{\pi_l}, \; \eta = \frac{w}{|A|}. \tag{8}$$

Here, $w$ is the variable that determines the extent of uniform random exploration in the action space. This presents a very challenging problem to learn a suitable belief representation for such agents that can be utilized by a principal to guide its intervention policy. In Section 6, we exclude EXP3 from Table 1 since it is primarily designed for an adversarial bandit setup, whereas we do not consider an agent to have such biases under our current problem formulation.

| $\beta$ | UCB | $\epsilon$-greedy | $\epsilon$ |
|---|---|---|---|
| 0.17 | 33 (0) | 33 (0) | 0.10 |
| 0.27 | 47 (0) | 47 (4) | 0.20 |
| 0.42 | 70 (0) | 68 (9) | 0.30 |
| 0.50 | 80 (0) | 81 (3) | 0.40 |
| 0.67 | 99 (0) | 99 (1) | 0.50 |

Table 4: **Experiment design choice.** Frequency of agent selecting $a_t \neq \arg\max_a r_a$ with UCB and $\epsilon$-greedy algorithms on the same set of base rewards (without any intervention) with a horizon $T = 200$, averaged across 3 random seeds.

### B.2 CHARACTERIZING THE DISTRIBUTION SHIFT IN OUR EVALUATION SETUP

Bandit agents having the same base reward $r$ make different explore-exploit decisions depending on their algorithm (eg. UCB, $\epsilon$-greedy) and also their prior observations. In Section 6, we consider agents with the same set of base rewards, but following different bandit algorithms. Both UCB and $\epsilon$-greedy have tunable parameters that determine their explore-exploit tradeoff. In order to measure the robustness of the learnt principal policy to different agent behavior (leading to different levels of non-stationarity in the principal's environment between training and test agents), we vary the amount of exploration performed by the agent by varying the respective parameters: $\beta$ for the UCB agent and $\epsilon$ for the $\epsilon$-greedy agent. Table 4 shows the average (and standard error) frequency of exploration by the agents for our choices of $\beta$ and $\epsilon$ in Section 6. We vary $\beta$ and $\epsilon$ such that they are pairwise comparable in Table 1 and would lead to similar change in exploration frequency for both UCB and $\epsilon$-greedy agents. In other words, following Table 1, a principal trained with UCB agents having $\beta = 0.17$ when evaluated with UCB agents having $\beta \in \{0.17, 0.27, 0.42, 0.50, 0.67\}$ will encounter a similar shift in the agent's exploration frequency as in the case of training with $\epsilon$-greedy agents with $\epsilon = 0.1$ and evaluating on $\epsilon$-greedy agents having $\epsilon \in \{0.1, 0.2, 0.3, 0.4, 0.5\}$. In that case, the difference in achieved scores between the UCB and $\epsilon$-greedy agents can be attributed to the way in which they distribute their exploratory actions: UCB agent being more optimistic focuses most of its exploration at the beginning of an episode, whereas the $\epsilon$-greedy agent is more stochastic with uniform random exploration throughout.

### B.3 DESCRIPTION OF BASELINES

We now describe the details of our evaluated baselines in Section 6 along with their variations that assume access to an agent state oracle.

**Rule based mechanism with an agent state oracle (RB):** Given an oracle that correctly identifies the action $a_t$ to be taken by an agent in the next time step, a simple rule based approach is for the principal to intervene at time $t$ when $a_t \neq a^*$. We assume that the principal always intervenes with a fixed incentive ($r' = 0.5$ or 1) and we compute the principal's maximum possible score. Note that this is not a realistic solution for the principal since it is impractical to expect the availability of such an oracle, especially for out of distribution test agents.

**Model-free learning based mechanism:** In this framework, we assume that the planner has a recurrent intervention policy that outputs a distribution over interventions $a_t^p \sim \pi_\theta^p\left(a_t^p | a_{t-1}^i, a_{t-1}^p, h_{t-1}^p\right)$, conditioned on the planner's intervention and observed agent action at $t-1$. The policy network is trained using REINFORCE for the MF-RL baseline and using MAML for the MF-MAML baseline.

**Learning based mechanism with an agent state oracle:** In this setting, the principal learns a recurrent intervention policy that outputs a distribution over interventions $a_t^p \sim \pi_\theta^p\left(a_t^p | a_t^i, h_{t-1}^p\right)$ conditioned on the true agent action at time $t$ provided by an oracle. The policy network is trained using REINFORCE for the SB-RL baseline and MAML for the SB-MAML baseline.

**Learning based mechanism with a world model without meta-learning (WM-RL):** In this setting, we use our proposed recurrent world model with a recurrent intervention policy trained

| **Train on UCB,** $\beta = 0.17$ | Test on $\beta = 0.17$ | $\beta = 0.27$ | $\beta = 0.42$ | $\beta = 0.5$ | $\beta = 0.67$ |
|---|---|---|---|---|---|
| *No intervention* | 3 (0) | 5 (0) | 8 (0) | 10 (0) | 12 (0) |
| RB | 173 (0) | 166 (0) | 154 (0) | 146 (0) | 126 (0) |
| SB-RL | 168 (3) | 138 (27) | 128 (26) | 122 (24) | 107 (22) |
| SB-MAML | 169 (3) | 169 (1) | 155 (2) | 148 (1) | 128 (2) |
| **Train on $\epsilon$-greedy,** $\epsilon = 0.1$ | $\epsilon = 0.1$ | $\epsilon = 0.2$ | $\epsilon = 0.3$ | $\epsilon = 0.4$ | $\epsilon = 0.5$ |
| *No intervention* | 3 (0) | 4 (1) | 7 (0) | 9 (1) | 11 (0) |
| RB | 156 (3) | 130 (1) | 105 (4) | 87 (4) | 62 (6) |
| SB-RL | 148 (2) | 119 (3) | 87 (4) | 75 (6) | 50 (2) |
| SB-MAML | 152 (1) | 126 (2) | 105 (3) | 66 (3) | 30 (9) |
| **Train on UCB,** $\beta = 0.67$ | $\beta = 0.17$ | $\beta = 0.27$ | $\beta = 0.42$ | $\beta = 0.5$ | $\beta = 0.67$ |
| *No intervention* | 3 (0) | 5 (0) | 8 (0) | 10 (0) | 12 (0) |
| RB | 173 (0) | 166 (0) | 154 (0) | 146 (0) | 126 (0) |
| SB-RL | 166 (3) | 163 (2) | 150 (3) | 146 (2) | 128 (2) |
| SB-MAML | 173 (1) | 170 (0) | 159 (0) | 152 (0) | 133 (0) |
| **Train on $\epsilon$-greedy,** $\epsilon = 0.5$ | $\epsilon = 0.1$ | $\epsilon = 0.2$ | $\epsilon = 0.3$ | $\epsilon = 0.4$ | $\epsilon = 0.5$ |
| *No intervention* | 3 (0) | 4 (1) | 7 (0) | 9 (1) | 11 (0) |
| RB | 156 (3) | 130 (1) | 105 (4) | 87 (4) | 62 (6) |
| SB-RL | 49 (46) | 51 (35) | 64 (29) | 61 (15) | 28 (17) |
| SB-MAML | 93 (45) | 62 (32) | 32 (13) | 58 (25) | 24 (17) |

Table 5: **Principal (with oracle agent state input) scores across 3 random seeds.** These baselines are not applicable in practice since they cheat by assuming access to an oracle that always informs them of the agent's next action. We include them here as a form of standardization with respect to a (perfect) system that does not face the challenges of partial observability or out-of-distribution generalization for mechanism design.

using REINFORCE. Here, the policy network outputs a distribution over interventions $a_t^p \sim \pi_\theta^p \left( a_t^p | \hat{a}_t^i, h_{t-1}^p \right)$ where $\hat{a}_t^i = \arg\max_a \hat{\pi}_\omega \left( a_t^i | a_{t-1}^i, a_{t-1}^p, h_{t-1}^i \right)$.

We would like to highlight an implementation detail in our baselines indicated 'RL' in Section 6. Since we evaluate our learnt principal policy in the $K$-shot adaptation setting which is common in the meta-learning literature, we ensure that the principal policies that are not meta-trained are also allowed to $K$-shot adapt at test time. This means that the 'RL' policies are also updated at test time, before evaluation, using $K$ rounds of principal-agent interactions. This is contrast to Section 4 where 'RL' was trained from scratch during test time adaptation. It further shows that even with pre-training (on the same set of train agents as used by 'MAML'), standard policy gradient update does not lead to test time $K$-shot adaptation on test agents.

In Table 5, we compare the test time scores for the principal policy having access to a state based oracle. We observe that overall, the meta-trained principal policy (SB-MAML) achieves a higher score even with distribution shift across different bandit algorithms and different levels of exploration, compared to the SB-RL baseline. The rule based baseline also shows strong performance but we note its scores do not reflect adaptation to distribution shift. However, none of these baselines that assume the principal has access to an oracle that correctly predicts the agent's action at the next time step are realistic. We can only treat the scores in Table 5 as gold standards in a perfect system that does not account for the challenges faced by a principal in practice.

**Training details.** In Section 4, the principal policy $\pi^p$ is a fully connected neural network (MLP) with one hidden layer and ReLU activation. Given an (noisy) observed value of the agent type as input, it predicts the probability of intervention: $\pi_t^p$. The principal's action at time $t$ is $a_t^p \sim$ Bern $(\pi_t^p)$.

For the 'RL' principal, it is trained on the test agents starting from scratch over $K$ episodes before evaluation. For the MAML principal, it is meta-trained to learn an initial parameterization with a different set of training agents and evaluated with $K$-shot adaptation on the test agents.

In Section 6, the recurrent world model and policy networks are GRUs with 2 layers and hidden state dimension 128. For meta-training, the inner gradient update loop uses SGD optimizer with a learning rate of $7 \times 10^{-4}$ whereas the meta-update step uses Adam with a learning rate of 0.001. The world model is trained only with the set of training agents, it is not adapted at test time: only the policy network is $K$-shot adapted.

---

**Algorithm 2** MERMAIDE ($K$-shot Adaptation)

---

1: Initialize principal with trained parameters ($\theta_{\text{meta}}, \omega_{\text{train}}$), and hidden states $h_0^i, h_0^p$.
2: **for** agents (tasks) $i = 1, \ldots, n_{\text{test}}$ **do**
3:     Initialize agent: ($\mu^i, \pi_0^i$), task specific intervention policy parameter $\theta\left(\tau_0^i\right) = \theta_{\text{meta}}$.
4:     **for** $k = 1, \ldots, K$ **do**                                         $\triangleright$ Inner loop for $K$ episodes.
5:         **for** time t $= 1, \ldots, T$ **do**          $\triangleright$ For each episode with $T$ principal-agent interactions
6:             Predict $\hat{a}_t^i = \arg\max_{a_t^i} \hat{\pi}_{\omega_{\text{train}}}\left(a_t^i | a_{t-1}^i, a_{t-1}^p, h_{t-1}^i\right)$ using the world model.
7:             Intervention: $\tilde{\mu}^i = \mu^i + a_t^p, \quad a_t^p \sim \pi_{\theta\left(\tau_k^i\right)}^p\left(a_t^p | a_{t-1}^i, a_{t-1}^p, \hat{a}_t^i, h_{t-1}^p\right)$.
8:             Agent acts: $a_t^i \sim \pi_t^i$ and receives reward $r_t^i \sim \mathcal{N}\left(\tilde{\mu}^i, \sigma^2\right)$. $\pi_t^i \mapsto \pi_{t+1}^i$.
9:         Locally update $\theta\left(\tau_k^i\right) \mapsto \theta\left(\tau_{k+1}^i\right)$.                $\triangleright$ Using REINFORCE.
10:     **for** $t = 1, \ldots, T$ **do**                               $\triangleright$ Rollout for evaluation
11:         Predict $\hat{a}_t^i = \arg\max_{a_t^i} \hat{\pi}_{\omega_{\text{train}}}\left(a_t^i | a_{t-1}^i, a_{t-1}^p, h_{t-1}^i\right)$ using the world model.
12:         Intervention: $\tilde{\mu}^i = \mu^i + a_t^p, \quad a_t^p \sim \pi_{\theta\left(\tau_K^i\right)}^p\left(a_t^p | a_{t-1}^i, a_{t-1}^p, \hat{a}_t^i, h_{t-1}^p\right)$.
13:         Agent acts: $a_t^i \sim \pi_t^i$, receives reward $r_t^i \sim \mathcal{N}\left(\tilde{\mu}^i, \sigma^2\right)$. Updates $\pi_t^i \mapsto \pi_{t+1}^i$.
14:         Update principal's score.

---

## B.4   Overview of $K$-shot adaptation with MERMAIDE:

Algorithm 2 outlines our framework for $K$-shot adaptation of the meta-trained principal to test agents. In our experiments, $K = 1$.

| Train on UCB, $\beta = 0.17$ | Test on $\beta = 0.17$ | $\beta = 0.27$ | $\beta = 0.42$ | $\beta = 0.5$ | $\beta = 0.67$ |
|---|---|---|---|---|---|
| *No intervention* | 3 (0) | 5 (0) | 8 (0) | 10 (0) | 12 (0) |
| MF-RL | 119 (2) | 109 (2) | 98 (2) | 90 (2) | 77 (1) |
| MF-MAML | 133 (2) | 125 (3) | 107 (1) | 97 (1) | 77 (0) |
| WM-RL | 123 (7) | 112 (6) | 100 (4) | 92 (2) | 75 (1) |
| MERMAIDE (ours) | **154 (2)** | **141 (1)** | **115 (1)** | **103 (0)** | **80 (1)** |
| MERMAIDE (ours) - 2nd set | 144 (3) | 133 (3) | 122 (2) | 108 (2) | 81 (2) |
| MERMAIDE ($K = 0$) | 148 (2) | 138 (1) | 120 (1) | 103 (2) | 89 (1) |
| WM-RL ($K = 0$) | - | 109 (1) | 92 (-) | - | - |
| **Train on $\epsilon$-greedy, $\epsilon = 0.1$** | $\epsilon = 0.1$ | $\epsilon = 0.2$ | $\epsilon = 0.3$ | $\epsilon = 0.4$ | $\epsilon = 0.5$ |
| *No intervention* | 3 (0) | 4 (1) | 7 (0) | 9 (1) | 11 (0) |
| MF-RL | 115 (5) | 94 (4) | 54 (19) | 39 (6) | 22 (9) |
| MF-MAML | 122 (4) | 97 (3) | 58 (5) | 40 (2) | 12 (1) |
| WM-RL | 115 (4) | 94 (5) | 70 (1) | 55 (3) | **38 (1)** |
| MERMAIDE (ours) | **134 (1)** | **108 (1)** | **85 (1)** | **57 (7)** | 29 (1) |
| MERMAIDE (ours) - 2nd set | 132 (3) | 111 (3) | 89 (1) | 68 (1) | 45 (1) |
| MERMAIDE ($K = 0$) | 133 (2) | 109 (3) | 86 (2) | 65 (3) | 37 (1) |
| **Train on UCB, $\beta = 0.67$** | $\beta = 0.17$ | $\beta = 0.27$ | $\beta = 0.42$ | $\beta = 0.5$ | $\beta = 0.67$ |
| *No intervention* | 3 (0) | 5 (0) | 8 (0) | 10 (0) | 12 (0) |
| MF-RL | 103 (3) | 101 (3) | 92 (2) | 85 (1) | 74 (1) |
| MF-MAML | 124 (2) | 116 (1) | 102 (1) | 94 (1) | 80 (1) |
| WM-RL | 100 (4) | 89 (0) | 85 (1) | 85 (1) | 74 (0) |
| MERMAIDE (ours) | **131 (2)** | **125 (2)** | **109 (1)** | **101 (1)** | **85 (1)** |
| MERMAIDE (ours) - 2nd set | 119 (7) | 118 (5) | 110 (3) | 104 (3) | 87 (2) |
| MERMAIDE ($K = 0$) | 115 (3) | 114 (3) | 103 (4) | 100 (3) | 89 (3) |
| WM-RL ($K = 0$) | 104 (8) | 90 (-) | - | 69 (1) | - |
| **Train on $\epsilon$-greedy, $\epsilon = 0.5$** | $\epsilon = 0.1$ | $\epsilon = 0.2$ | $\epsilon = 0.3$ | $\epsilon = 0.4$ | $\epsilon = 0.5$ |
| *No intervention* | 3 (0) | 4 (1) | 7 (0) | 9 (1) | 11 (0) |
| MF-RL | 4 (5) | 2 (3) | 5 (0) | 11 (5) | 7 (1) |
| MF-MAML | 2 (0) | 4 (0) | 6 (0) | 8 (1) | 11 (1) |
| WM-RL | 102 (6) | 79 (10) | 68 (3) | 47 (1) | 30 (2) |
| MERMAIDE (ours) | 87 (42) | **102 (3)** | **78 (6)** | **69 (1)** | **46 (2)** |
| MERMAIDE (ours) - 2nd set | - | 89 (17) | 65 (12) | 47 (20) | 20 (15) |
| MERMAIDE ($K = 0$) | 113 (20) | 85 (15) | 71 (16) | 48 (14) | 21 (15) |

Table 6: **Test-time principal mean and standard error scores.** Left column: Principal's algorithm (e.g., MERMAIDE), training agent type (e.g., UCB with $\beta = 0.17$). Other columns: Test-time scores on agents with the same algorithm, but different hyperparameters. Gold-colored values represent testing with a principal that is *not updated during test-time, i.e., $K = 0$-shot generalization.* We see that the principal can perform on par when trained on $\beta = 0.17$ and $\epsilon = 0.1$, but that 0-shot generalization does not work so well when the principal was trained on more exploratory hyperparameter values, i.e., $\beta = 0.67$ and $\epsilon = 0.5$. To compare the levels of exploration between different hyperparameter settings, please refer to Appendix B.2 and Table 4. All results are based on 3 random seeds. Note that the results in blue use 3 random seeds with the same settings as the rows above; as such, there are two sets of 3 random seeds for MERMAIDE. We see that the results are similar between the two sets of 3 random seeds.

## C  ADDITIONAL EXPERIMENTAL RESULTS WITH BANDIT AGENTS

**Additional seeds for Table 1.**  We ran the same set of experiments for MERMAIDE from Section 6 with 3 additional results. At the time of the rebuttal deadline, some of the runs have not converged so we are reporting these additional results in the row marked in blue in Table 7. We will update these with final converged values and more seeds if requested for the camera ready version if the paper is accepted. On the basis of our current results, we do not expect a significant variation from the values originally reported in Table 1 even with more seeds.

**Zero-shot evaluation results with MERMAIDE.**  As requested by reviewer 4J8Y, we evaluated MERMAIDE in the zero-shot setting i.e. the policy is not updated at meta-test time ($K = 0$). Table 7 shows the preliminary results for these experiments in gold. Some of these experiments have not coverged yet, but we do not expect a lot of improvement compared to the reported scores. We see that the principal can perform on par when trained on $\beta = 0.17$ and $\epsilon = 0.1$, but that 0-shot generalization does not work so well when the principal was trained on more exploratory hyperparameter values, i.e., $\beta = 0.67$ and $\epsilon = 0.5$. To compare the levels of exploration between different hyperparameter settings, please refer to Appendix B.2 and Table 4.

| Train on UCB, $\beta = 0.42$ | Test on $\epsilon = 0.1$ | $\epsilon = 0.2$ | $\epsilon = 0.3$ | $\epsilon = 0.4$ | $\epsilon = 0.5$ |
|---|---|---|---|---|---|
| *No intervention* | 3 (0) | 4 (1) | 7 (0) | 9 (1) | 11 (0) |
| WM-RL | 91 (4) | 62 (8) | **68 (1)** | **28 (4)** | - |
| MERMAIDE (ours) | **103 (1)** | **67 (2)** | 30 (2) | 8 (1) | - |
| **Train on $\epsilon$-greedy, $\epsilon = 0.3$** | Test on $\beta = 0.17$ | $\beta = 0.27$ | $\beta = 0.42$ | $\beta = 0.5$ | $\beta = 0.67$ |
| *No intervention* | 3 (0) | 5 (0) | 8 (0) | 10 (0) | 12 (0) |
| WM-RL | 127 (7) | 95 (2) | 80 (5) | 80 (5) | 61 (4) |
| MERMAIDE (ours) | **138 (2)** | **102 (6)** | **116 (2)** | **96 (5)** | **77 (2)** |

Table 7: **Test-time principal mean and standard error scores across 3 random seeds, $K = 0$.** Left column: Principal's algorithm (e.g., MERMAIDE), training agent type (e.g., UCB with $\beta = 0.42$). Other columns: Test-time scores on agents with different algorithm and different hyperparameters. We see that MERMAIDE generalizes well when tested on agents that explore the same amount or less than the train-time agents. For clarity, note that higher $\beta$ and $\epsilon$ lead to more exploration. More generally, a principal that is trained on a stochastic agent generalizes well to an equal or less stochastic agent, e.g., training on $\epsilon = 0.3$ and testing on UCB with $\beta = 0.5, 0.67$; note that the behavior of UCB is less stochastic than $\epsilon$-greedy.

**Cross algorithm evaluation.** Table 7 indicates the mean and standard error scores for evaluation in the $K = 0$-shot generalization setting when the training agent and test agents are of different types. Note that the behavior of UCB agents is less stochastic than $\epsilon$-greedy agents. We observe that when trained with UCB agents, MERMAIDE outperforms WM-RL for generalizing to $\epsilon$-greedy agents that have a lower exploration coefficient $\epsilon = 0.1$ or $0.2$. In contrast, when trained with $\epsilon$-greedy agents, MERMAIDE outperforms WM-RL for generalizing to UCB agents with both higher and lower levels of exploration. More generally, a meta-learning principal that is trained on a stochastic agent generalizes well to an equal or less stochastic agent in the zero shot setting.

