# OpenReview forum: "MERMADE: $K$-shot Robust Adaptive Mechanism Design via Model-Based Meta-Learning"
_ICLR.cc/2023/Conference — Submitted to ICLR 2023_

### Official Review · Reviewer_2WWc · 2022-10-17

**Confidence:** 3
**Correctness:** 3
**Technical Novelty And Significance:** 2
**Empirical Novelty And Significance:** 2
**Recommendation:** 5

**Clarity, Quality, Novelty And Reproducibility:**

I cannot comment on the originality of this work as I expect that it has exceptional similarity to work in Inverse Reinforcement Learning, of which I am not well-read. If it is novel in that area, then comparing it to [1] there is a novelty in looking at having the mechanism designer incur a cost for implementation.

The overall clarity, quality, and reproducibility of this work is better than average; however, I would not expect the results to be replicated without the source code, and would encourage the authors to release it.

[1] Fu, et al.. Evaluating Strategic Structures in Multi-Agent Inverse Reinforcement Learning. JAIR, 2021.

**Strength And Weaknesses:**

**Strengths**
 - The exposition following their framing of intervened rewards and the planner was very clear.
 - Inclusion of the per-opponent generalization in Table 1 provides better insights into generalization, and the contribution of the model-based additions to their algorithm.
 - Mechanism Design is an understudied field within multiagent reinforcement learning methods and the area needs more attention.

**Weaknesses**
 - Meta-learning appears to be an exceptionally complex approach for mechanism design in the provided experiments. It would embolden the author's work to either show analytic solutions and how their algorithm offers an advantage, and/or demonstrate their algorithm on a game where we cannot compute analytic or Bayesian solutions.
 - The authors only investigate a single-agent environment and analyze repeated interactions between the agent's behavior and implementation of mechanisms derived from a co-learner independently. This is strange for a piece on mechanism design, because the incentives of both "players" (the agent and mechanism designer) are aligned. There is no strategic component present such as an agent not being truthful about their policy. I am curious if the authors have invested applying this method to general-sum games with >1 player  and if their mechanism selects equilibrium that are strategy-proof, efficient, or balanced?
 - Furthermore, as this work focuses on single-agent environments, this work is effectively in inverse reinforcement learning. I was hoping the authors could comment on how their algorithm differs from the existing work in this space? I would also ask that they include this discussion in their paper. They should also include baselines from this field in this work.
 - A major point in their claims is that their algorithm can quickly adapt to unseen test agents. As a result, it's surprising to me that the results all assume that the train and test agents are using the same underlying algorithm (with different exploration coefficients). Could the authors comment on the generalization performance of their algorithm compared to baselines _across_ algorithms? Moreover, comment on the diversity of the population exhibited by the coefficients are chosen, it's not clear to me if beta=0.17 or beta=0.27 create meaningfully different behaviors in such a simple game.
 - I found their usage of test/train/meta-test/meta-train and its relation with varying k-shot regimes to be confusing. Perhaps that's just me, but for example ".... adapted quickly on a single test-time trajectory", is ambiguous as if the adaptation must occur within the trajectory, or if the adaptation is conditioned on a single trajectory. Moreover, if either is the problem setting, then why is there commentary about K-shots?
 - The name of the algorithm is never explained?

**Summary Of The Paper:**

Mechanism design studies the problem of constructing a game that elicits a desired behavior from its' players. Often players adapt their policy with feedback through a learning algorithm to improve their future performance. This raises the question of how to design a mechanism that elicits a specified behavior while being agnostic to its' players' learning algorithms. This manuscript proposes an algorithm that learns to modify a single-agent player's behavior by altering said agent's reward function.

The authors claim the following contributions:
 - A model-based meta-learning mechanism designing algorithm for few-shot games with unknown agents.
 - Empirically demonstrate their algorithm on single-stage matrix games with perfect and noisy observation.
 - Empirically demonstrate their algorithms on multi-stage matrix games with learning agents.


**Summary Of The Review:**

Overall, I'm excited to see work in this area, but the lack of discussion of inverse reinforcement learning makes it hard to understand where this paper fits into the literature. Moreover, without IRL baselines and narrative, it's unclear if this work contributes a new idea or one that is competitive in this domain. As these are major scientific concerns with this work I would not recommend publishing this work in its current state, but encourage the author's to continue to pursue this topic.

---

> ### Author Response · Authors · 2022-11-12
> **Response to reviewer 2WWc**
>
> We thank you for your time and thoughtful review. We address your primary concerns as follows:
>
> 1. *“..show analytic solutions and how their algorithm offers an advantage, and/or demonstrate their algorithm on a game where we cannot compute analytic or Bayesian solutions.”* : In Section 4, we show that meta-training helps achieve the expected Stackelberg equilibrium in a simple matrix game setting, compared to first order policy optimization using REINFORCE. Next, in Section 6, we demonstrate the superior empirical performance of MERMADE over competing baselines for mechanism design with bandit agents having different types. In addition to our response above about the NP-hardness of the mechanism design problem (please refer to our comments about the problem setup), we would like to reiterate that our setup with adaptive bandit agents and incomplete agent information already presents a non-trivial problem for learning-based mechanism design.
>
> 2. *“...the incentives of both "players" (the agent and mechanism designer) are aligned.”* : We would like to clarify that the incentives of the mechanism designer and the agent are **not** always aligned. In the MAB setting, the agent can prefer a different action than the planner’s preferred action i.e. $\tilde{a} = \arg\max_a r_a$ (the agent’s preferred action without any intervention) may not always match $a^*$ (the planner’s preferred action).  In this case, the goal of the mechanism designer is to intervene appropriately so that the agent can be incentivized to (learn to) prefer $a^*$ instead of $\tilde{a}$.
>
> 3. *“...this work is effectively in inverse reinforcement learning”* : We disagree with this characterization and briefly describe what differentiates our setting from prior work in inverse reinforcement learning in section 2 in the main paper. To reiterate, inverse RL algorithms aim to estimate a reward function for the observed demonstrations from an expert policy that behaves optimally for that reward throughout the demonstration. This is different from our adaptive mechanism design setup in the following ways:
>     - In our problem setting, the agents are themselves learning an optimal policy during each episode. So the mechanism designer does not have access to expert demonstrations, unlike IRL. The agents’ base rewards, internal states or learning strategies are unknown to the mechanism designer and it only observes their actions.
>     - Unlike IRL, the mechanism designer’s goal is to determine a cost-aware intervention strategy so that the agent can be incentivized to (learn to) prefer $a^*$ instead of $\tilde{a}$. The mechanism designer should therefore be able to reason about both the agent’s base rewards and learning algorithm to decide when and how to intervene. This is a higher order problem setting than only reward learning, therefore we do not compare to IRL baselines in this work.
>
> 4. *Generalization performance of MERMADE compared to baselines across different (bandit) agent algorithms* : Following our discussion in Section 6, recall that the nature of the bandit algorithm (UCB or $\epsilon$-greedy) determines the temporal (within episode) distribution of exploration vs exploitation for approximately the same number of total exploratory actions in an episode. We therefore observed that the world model and the meta-learned planner policy are independently advantageous under different test conditions for MERMADE.
>     - When train agents are $\epsilon$-greedy and test agents follow UCB, for equivalent $\epsilon$ and $\beta$ i.e. approximately same number of total exploratory actions in an episode, MERMADE is expected to outperform other baselines because compared to $\epsilon$-greedy, the UCB agent is less stochastic and explores mostly at the start of an episode. This train-test setting is therefore similar to that of “Train on $\epsilon$-greedy, $\epsilon=0.5$” in Table 1.
>     - On the other hand, when train agents follow UCB and test agents are $\epsilon$-greedy, the test agent being more exploratory and stochastic, the trained world model would provide a better prior for the next-agent-action and together with the MAML update,  MERMADE is expected to outperform baselines in terms of OOD generalization in that setting.  This train-test setting is therefore similar to that of “Train on UCB, $\beta=0.17$” in Table 1.

---

> > ### Author Response · Authors · 2022-11-12
> > **Continued response to reviewer 2WWc**
> >
> > 5. *“...comment on the diversity of the population exhibited by the coefficients are chosen”* :  We would like to refer to Section B.2 and Table 4 in the Appendix. Table 4 shows that $\beta=0.17$ and $\beta=0.67$ (as well as $\epsilon=0.1$ and $\epsilon=0.5$) have a significant difference in the number of times the agent explores actions that do not result in the maximum reward, given the same base rewards and without any external intervention. As we have described in Section 6, since bandits learn with sequential feedback and the bandit algorithm affects how the exploration actions are distributed in an episode, our range of parameters $\epsilon$ and $\beta$ were chosen to reflect a reasonable variation in the agent behavior between training and test time.
> >
> > 6. We would like to emphasize that K-shot has been used only in the context of the test time adaptation setting i.e. the planner can interact with the test agent over $K$-episodes to update the learned policy weights using MAML and this test-time adapted mechanism designer is then evaluated over a final $K+1$-th episode (see section B.4, Algorithm 2 in the appendix). We have explicitly (described and) used $K=1$ and to make this clearer, we can appropriately replace $K$-shot with 1-shot in the paper if recommended. *".... adapted quickly on a single test-time trajectory"* implies that the adaptation is conditioned on a single observed trajectory ($K=1$) at test time.
> >
> > 7. MERMADE stands for “**Me**ta-lea**r**ning for **M**odel-based **A**daptive Mechanism **De**sign”. We will include this in the main paper.
> >
> > 8. We will make the code publicly available once the paper is accepted.

---

> > > ### Comment · Reviewer_2WWc · 2022-11-17
> > > **Response**
> > >
> > > Thank you for taking the time to reply. I found the edits to the paper generally agreeable and that it has greatly improved the piece. I have increased my "score" correspondingly.
> > >
> > > I agree in hindsight that IRL was maybe not the most appropriate characterization. Much like reviewer C92e, I found the casting of this work as a mechanism design problem to be a misalignment with the true scope of the work. The main throughline of both of our reviews is that this work has a similar problem framing as other problems that focus on the strategic modification of reward functions, whether that strategy is a curricula, a strategic/heuristic shaper, etc.. Re-opening the paper I see that it has largely changed reflecting a more appropriate scoping. With these two thoughts in mind, I still maintain my initial position that the work could be largely improved with additional baselines focused on the reward-modification component as opposed to the response calculation component.
> > >
> > > Given the magnitude of changes that have occurred I would generally recommend that the paper go through another reviewing cycle. For example: there are still references to MERMADE, despite the algorithm being renamed. I expect there are other portions of the text that need a similar revisiting.

---

> > > > ### Author Response · Authors · 2022-11-18
> > > > **Thank you for your response!**
> > > >
> > > > We have replaced the remaining mentions of MERMADE in the appendix, thank you for pointing it out. We have appropriately updated portions of the text following the recommendations of reviewer C92e to better describe our problem setup as learning to align adaptive learners. Using the pdf comparison tool available on Openreview, we have generated this comparison of two versions of our uploaded submission and you can access it on this link: https://api.draftable.com/v1/comparisons/viewer/yVmSPr/pyAiQsQcspfA?valid_until=1668739830&signature=7d489cedf4eaa4ac1f97510ee98fdc656c7d611644e802798085017d59885716&wait. We are also attaching the annotated version of the comparison generated using this tool to the supplementary material and updating the submission here. You will notice that, consistent with our response to reviewer C92e, our edits are primarily limited to change in title, some additional sentences to make the message consistent in the Abstract and Introduction and replacement of MERMADE $\rightarrow$ LEAAL and planner $\rightarrow$ principal. We have taken care to ensure that the text is consistent throughout while accounting for the reviewer's suggestion. *Please let us know if this annotated comparison is satisfactory to easily understand the textual differences in the current version of our paper.*
> > > >
> > > > Regarding additional baselines, we would appreciate it if you could please point us to specific baselines that you think we are missing in the current work.

---

> > > > ### Author Response · Authors · 2022-11-19
> > > > **We have added additional results to the appendix**
> > > >
> > > > Note that in the updated version, we have renamed our algorithm to MERMAIDE (we apologize for the name change again).
> > > >
> > > > **Cross algorithm results:** We have added the results for the zero-shot evaluation in the cross-algorithm setting in Table 7 in Appendix C in the updated submission. We see that MERMAIDE generalizes well when tested on agents that explore the same amount or less than the train-time agents. For clarity, note that higher $\beta$ and $\epsilon$ lead to more exploration. More generally, a principal that is trained on a stochastic agent generalizes well to an equal or less stochastic agent, e.g., training on $\epsilon = 0.3$ and testing on UCB with $\beta = 0.5, 0.67$; note that the behavior of UCB is less stochastic than $\epsilon$-greedy.
> > > >
> > > > **Additional seeds for MERMAIDE, $K=1$:** We have also included 3 additional seeds for MERMAIDE and report them in blue in Table 6 in Appendix C. Since some of these experiments are yet to converge, we have reported them separately along with our original results. We observe that we get similar mean and standard error for the scores compared to our previously reported values, thereby showing the stability of our training process and reproducibility in the reported results.
> > > >
> > > > **Added results for MERMAIDE, $K=0$:** Moreover, we have also included some preliminary results for evaluation of MERMAIDE in the zero-shot setting in Table 6 in Appendix C (rows colored in gold). We see that the principal can perform on par when trained on $\beta = 0.17$ and $\epsilon = 0.1$, but that 0-shot generalization does not work so well when the principal was trained on more exploratory hyperparameter values, i.e., $\beta=0.67$ and $\epsilon=0.5$.  To compare the levels of exploration between different hyperparameter settings, please refer to Appendix B.2 and Table 4.
> > > >
> > > > *We hope this addresses your comment regarding additional experimental evaluations in our updated paper.*

---

### Official Review · Reviewer_C92e · 2022-10-24

**Confidence:** 4
**Correctness:** 3
**Technical Novelty And Significance:** 3
**Empirical Novelty And Significance:** 3
**Recommendation:** 6

**Clarity, Quality, Novelty And Reproducibility:**

Reproducibility is lacking due to no code being provided and the notorious difficulty of reproducing (META)-RL methods.
Evaluation is mismatched with the paper claim as described in weaknesses, and I am also missing error bars (unless the parenthesis denote std, which should be noted in table caption). Also, 3 seeds of a single META algorithm is too little, unless I'm missing a an explaination that this is justified.
Clarity is well done otherwise.

**Strength And Weaknesses:**

Strengths:

- I think this is an interesting first step towards learning best mechanism design. Nudging two agents playing prisoners dilemma towards cooperating and being able to identify the minimal number of interventions *dynamically* by meta-adapting is an intuitive "minimal" mechanism design
- the experiments are well thought out, explained and reasoned for
- it is interesting to see that the model based mechanism can learn to recognise agent behaviour and

Weaknesses:

- the biggest problem I have with the paper is that as of now, it isn't really mechanism design, it's simply curriculum learning for bandit algorithms. For it to truly be mechanism design, I would expect there to be a *game* between no-regret learners (as above, nudging *two* players, making for a three player game). As of right now, one uncharitable way of characterising the paper would be "empirical evaluation that MAML vs no-regret bandit algorithm games converge" (which, I must emphasise, is still a fine and nontrivial contribution). I think this can be alleviated by discussing the limitations the choice to focus on single agent poses, or even better, by evaluating the system in the 2-players, 1 planner setting, with "no planner" being a natural baseline. To be honest, I expect this to be *much* more difficult, but even a negative result would strengthen the contribution of this paper by establishing that a single adaptive algorithm (albeit simple) is "steerable" while a game might not be anymore

**Summary Of The Paper:**

The paper presents a MAML approach to meta-training a model based RL policy which will learn to "nudge" no-regret bandit learning algorithms into cooperating in a single-player prisonners dilemma. Experimental evaluation shows that the model based method allows "level-2" reasoning about the "level-1" decision and adaption rule of the bandit algorithm.

**Summary Of The Review:**

Overall, I'd say this paper *barely* misses the threshold to being accepted. If the additional experiment on 2 players can be added I think it would be a clear accept, if not it would depend on the quality of discussion on why this is justified.

---

Post-rebuttal edit: the reframing of narrowing of scope from "mechanism design" to "learning to align learners" done by the authors after our discussion and the additional cross-algorithm experiments made me change my score from 5 to 6

---

> ### Author Response · Authors · 2022-11-12
> **Response to reviewer C92e**
>
> We thank you for your time and thoughtful review. We address your primary concerns as follows:
>
> 1. We would like to point out that a minimal mechanism design framework does not necessarily involve the mechanism designer trying to influence the outcome of a 2-player game (please also refer to our comments about the problem setup above). For example, in the setting of sequential posted pricing [1], the mechanism designer interacts sequentially with each agent - one at a time, and there is no inter-agent competition or coordination.
> Even in our problem setting of learning based mechanism design with a single agent, it is non-trivial to design a mechanism because of the following reasons:
>     - The agents are themselves learners, trying to identify their own optimal actions among multiple feasible choices. Different agents have their own learning strategies which affects how they respond to the mechanism in the sequential setting (see Section 6 for more description of the challenges faced for mechanism design in the sequential bandit setting).
>     - The mechanism designer intends to incentivize the learning process of an agent so as to maximize the designer’s objective while minimizing the incurred cost of such incentivization. In the absence of complete information about the agent’s learning strategy and in an action space with multiple feasible choices for each agent, this presents a challenging learning and generalization problem with appreciable intra-episode and inter-episode non-stationarity.
>
> 2. Compared to curriculum learning, in our problem formulation, the goal of the mechanism designer is not to merely learn the explore-exploit algorithm being followed by the bandit agent. Instead, the mechanism designer’s policy decides when and how to intervene so that the bandit agent can be incentivized to align its preferred action with that of the mechanism designer, which is different from the agent’s preferred action. As a result of the interaction between the mechanism designer and the agent during each episode, the mechanism designer as well as the (learning) bandit agent adapt to the non-stationarity in their environments. This is in contrast to a typical teacher-student curriculum learning framework in which the teacher selects a fixed task for the student at the beginning of each episode, with a different task being selected across episodes.
>
> 3. We agree that mechanism design with a multi-agent game is a significantly challenging setting because of the following reasons:
>     - the agents are themselves learners and in a multi-agent system, the mechanism designer has to learn how the agents *jointly* learn based only on their observed actions
>     - there may not exist a unique equilibrium in a general sum multi-agent game, which adds to the difficulty of the learning problem for the mechanism designer
>     - the mechanism designer’s intervention may affect the agents differently and lead to the emergence of different cooperation or competition behavior among the agents - this further complicates the second order reasoning problem for the mechanism designer to decide when and how to intervene
>
>     So we leave the problem of mechanism design in a multi-agent game to future work. We have already discussed why our current single-agent setting presents a non-trivial mechanism design problem and MERMADE empirically shows the benefits of our model-based meta-learning approach over competing baselines for learning based mechanism design.
>
> 4. In Table 1, the parentheses indicate the standard error across 3 seeds and we will specify this in the caption. We are currently working to report results with 6 additional random seeds for MERMADE, but based on the mean and standard error values in Table 1, we do not expect a significant variance in performance.
>
> [1] Chawla, S., Hartline, J.D., Malec, D.L. and Sivan, B., 2010, June. Multi-parameter mechanism design and sequential posted pricing. In Proceedings of the forty-second ACM symposium on Theory of computing (pp. 311-320).

---

> > ### Comment · Reviewer_C92e · 2022-11-12
> > **Thank you for your answer**
> >
> > Thank you for pointing to these references. I agree that one *can* frame your algorithm as designing and adaptive randomized algorithm (the meta-learner performing the design, the initialisation of the trainer combined with REINFORCE at test time being the mechanism). However, one can also frame this as learning a good init of an RL algorithm playing a against a no-regret learning agent in a stationary game, and to me this seems like the better framing since
> >
> > 1. the WM-RL method seems to retain most of the benefits (in some cases even outperforming mermade)
> > 2. the *mechanism* (the RL polilcy)  is only encountering a fixed no-regret learner, meaning the dynamics of the game it needs to adapt to are fixed. While the opponents *actions* are not fixed, this standard when learning in games
> > 3. Both the SPM mechanism and the two-day auction (TDA) you refer to as examples have significant differences to your setting. Both fix mechanisms ahead of time that  will have to adapt within a single episode of play by conditioning (while you adapt the planner across multiple episodes, allowing the WM to estimate dynamics). Both mechanisms have as their challenge to find a setting that ensures incentives *independent* of the particular state the agent might be in, while the main challenge for the planner is to 1) estimate the exploration factor of a learning algorithm it was already trained on as part of the world model 2) use that estimated factor in it's WM to nudge the learner towards  a specific goal
> > 4. I am pretty sure (although not totally sure) you can rewrite the game you constructed  as a zero sum min-max game between finding the highest paying arm (for the agent) and another arm (for the planner) which is a much simpler problem than mechanism design.
> >
> > These above reasons make me describe the method as curricula learning -  more fairly, adaptive curricula learning. Indeed, I'd give  a higher rating if the paper was pitched as "Adaptive curricula learning" or "Learning to nudge no-regret learners" since then it would be adequately scoped. By framing it as "mechanism design", for me there should be at least *one part* which clearly distinguishes it against "simple" learning in zero-sum games.
> >
> > One possibility that occurred to me while writing this answer: what happens if you evaluate *cross-algorithm* or have the agent algorithm change between episodes? I think it would be even be fair to meta-train the planner on this (although the zero-shot performance might be more feasible given time constraints and also interesting), and it would remove part of my critique in that it move curriculum learning for a single algorithm to generalising curriculum learning which *is* difficult to distinguish from mechanism design for me.
> >
> > I hope my arguments appear consistent to you, even if you disagree. I still think it is a decent paper (and if n-player experiments/analysis and/or cross algorithm results are added, even a good paper!), but the above reasoning makes me retain my skepticism.

---

> > > ### Author Response · Authors · 2022-11-17
> > > **Thank you for your detailed response!**
> > >
> > >
> > > - We respectfully disagree with the characterization of our method as curriculum learning. Following [1], curriculum learning would imply a training strategy in which the training tasks are made incrementally more difficult with the end goal of achieving some optimal learning outcome. In contrast, in our problem setup, the planner’s reward intervention at any time step in an episode essentially **changes the task** for the bandit agent, incentivizing it to select a different action (compared to its intrinsic preferred action) to obtain the maximum reward.
> > >
> > > - We agree with your comment that this setting could be framed as learning to nudge no-regret learners and realize that using the term mechanism design in the title might imply a broader scope of problems. We have updated our paper to address this concern and modified some parts of the text to better describe our studied problem setup. *Please let us know if this addresses your concerns with the messaging and scoping of the paper.*
> > >     - Specifically, we have updated the title of the paper and the titles in sections 3 and 5 to better reflect the problem setup and our solution approach. Our proposed method has been renamed to LEAAL from MERMADE.
> > >     - We have replaced instances of the term “planner” with “principal” throughout the text.
> > >
> > >
> > > - We are working on adding the zero-shot evaluation results across algorithms to the paper, comparing the performance of WM-RL and our proposed method.
> > >
> > > - Multi-agent games present complex issues related to the existence of multiple equilibria in general-sum games, which introduces the equilibrium selection problem. This distracts from our objective of generalizing well to agents with unseen learning algorithms at training time. Hence, we believe the current scope is appropriate to validate the contribution of our work.
> > >
> > > [1] Bengio, Yoshua, et al. "Curriculum learning." Proceedings of the 26th annual international conference on machine learning. 2009.

---

> > > > ### Comment · Reviewer_C92e · 2022-11-17
> > > > **Thank you**
> > > >
> > > > Dear authors, first, thank you very much for engaging with me. Yes, the new framing fixes the biggest problem of the paper for me and I will update accordingly. Please share any preliminary results if the interaction period/time to modify the paper runs out before the full evaluation finishes, even partial results will be much appreciated (if you do not want to update the paper, feel free to share via e.g. imgur links). I agree that given the new re-scoping, it is sufficient to omit multi player games.
> > > >
> > > > ---
> > > >
> > > > Now, strictly to continue the discussion I'd like to respond to
> > > >
> > > > >We respectfully disagree with the characterization of our method as curriculum learning. Following [1], curriculum learning would imply a training strategy in which the training tasks are made incrementally more difficult with the end goal of achieving some optimal learning outcome. In contrast, in our problem setup, the planner’s reward intervention at any time step in an episode essentially changes the task for the bandit agent, incentivizing it to select a different action (compared to its intrinsic preferred action) to obtain the maximum reward.
> > > >
> > > > I agree with this criticism of the term curriculum learning given it's current technical usage. Aligning (as you have chosen) might be a better term, other options might be "automated reward shaping" or "reward engineering". In any case, for me the crucial difference to mechanism design is that you do not fix or tune a mechanism to create or regulate a game between players, but instead design a player that engages in a (very specific) game structure with a single player As I said, I appreciate the fact that the line between "designing a player" and "designing an adaptive mechanism" is fuzzy and if there had been a multi player component *or* a strict online nature to the mechanism *or* another way to cleanly delineate this from pre-training a player in a specific game, I think I would not have insisted so much on this. As it is, I think the reframing and rescoping is a distinction with a difference that I'm happy we could resolve.

---

> > > > > ### Author Response · Authors · 2022-11-18
> > > > > **Preliminary additional results**
> > > > >
> > > > > At the moment, we have a few results from the additional experiments we are running and we hope to have more by Nov 18 AoE before the deadline.
> > > > >
> > > > > For the cross algorithms, we have the following results for *zero-shot evaluation*:
> > > > > 1) Training with UCB, $\beta=0.42$ and test with $\epsilon$-greedy, $\epsilon=0.1$:
> > > > >
> > > > >    |Algorithm   |  Score |
> > > > >    | :---            |     ---:  |
> > > > >    | WM-RL     |   90 (4) |
> > > > >    | LEAAL      |   95 (7) |
> > > > >
> > > > > In this setting, the agent at test time is less exploratory than the training agent. So, we observe a smaller margin of performance in the zero-shot evaluation setting between meta-training a policy versus standard RL training, in the presence of a trained world model predicting the the next-agent-action.
> > > > >
> > > > > 2) Training with $\epsilon$-greedy, $\epsilon=0.3$ and test with UCB, $\beta=0.67$:
> > > > >
> > > > >     |Algorithm   |  Score |
> > > > >     | :---            |     ---:  |
> > > > >     | WM-RL     |  61 (4)  |
> > > > >     | LEAAL       | 78 (2)  |
> > > > >
> > > > > In this setting, the test agent has a higher tendency of exploration than the train agent and although the UCB agent is less stochastic than the $\epsilon$-greedy agent and explores mostly at the start of an episode, we observe a greater performance improvement over WM-RL by meta-training the intervention policy.

---

> > > > > > ### Author Response · Authors · 2022-11-19
> > > > > > **We have added the additional results to the appendix**
> > > > > >
> > > > > > Note that in the updated version, we have renamed our algorithm to MERMAIDE (we apologize for the name change again).
> > > > > >
> > > > > > **Cross algorithm results:** We have added the results for the zero-shot evaluation in the cross-algorithm setting in Table 7 in Appendix C in the updated submission. We see that MERMAIDE generalizes well when tested on agents that explore the same amount or less than the train-time agents. For clarity, note that higher $\beta$ and $\epsilon$ lead to more exploration. More generally, a principal that is trained on a stochastic agent generalizes well to an equal or less stochastic agent, e.g., training on $\epsilon = 0.3$ and testing on UCB with $\beta = 0.5, 0.67$; note that the behavior of UCB is less stochastic than $\epsilon$-greedy.
> > > > > >
> > > > > > **Additional seeds for MERMAIDE, $K=1$:** We have also included 3 additional seeds for MERMAIDE and report them in blue in Table 6 in Appendix C. Since some of these experiments are yet to converge, we have reported them separately along with our original results. We observe that we get similar mean and standard error for the scores compared to our previously reported values, thereby showing the stability of our training process and reproducibility in the reported results.
> > > > > >
> > > > > > **Added results for MERMAIDE, $K=0$:** Moreover, we have also included some preliminary results for evaluation of MERMAIDE in the zero-shot setting in Table 6 in Appendix C (rows colored in gold). We see that the principal can perform on par when trained on $\beta = 0.17$ and $\epsilon = 0.1$, but that 0-shot generalization does not work so well when the principal was trained on more exploratory hyperparameter values, i.e., $\beta=0.67$ and $\epsilon=0.5$.  To compare the levels of exploration between different hyperparameter settings, please refer to Appendix B.2 and Table 4.

---

### Official Review · Reviewer_9dtX · 2022-10-25

**Confidence:** 3
**Correctness:** 3
**Technical Novelty And Significance:** 3
**Empirical Novelty And Significance:** Not applicable
**Recommendation:** 5

**Clarity, Quality, Novelty And Reproducibility:**

Clarity: Lack of a clear graph to reflect the structure and novelty of the algorithm.
Quality: The charts are rough, and lack detailed instructions.
Novelty: The algorithm proposed in this paper combines model-based reinforcement learning and meta-learning. Compared with traditional methods, the designed mechanism has better adaptability.
Reproducibility: The author did not provide the source code, so I cannot confirm it


**Strength And Weaknesses:**

a)	Strength:
Different from the way of mechanism design in traditional game theory, this paper combines model-based and meta-learning theory in reinforcement learning to design a mechanism with good robustness and adaptability.
b)	Weaknesses:
In model-based reinforcement learning, the author lacks sufficient analysis of model error and policy shift and lacks a detailed description of how the model is established. The experimental results are not fully displayed graphically, and the paper is not very easy to understand. Experimental results do not have multiple sets of seeds to eliminate probabilistic errors.


**Summary Of The Paper:**

This paper proposed MERMADE, a deep RL approach to mechanism design that fuses model-based methods and gradient-based meta-learning methods to design a mechanism with fast adaptability. The authors analyze the one-shot adaptation performance of a meta-learned planner in a matrix game setting, under both perfect and noisy observations for the agent and the planner. They show that meta-training reliably finds solutions that one-shot adapt well, and characterize how the planner’s out-of-distribution performance depends on its observable information about the agent.

**Summary Of The Review:**

This paper proposes a mechanism design algorithm based on the model-based method and meta-learning, and learns the adaptive mechanism, but lacks theoretical analysis, insufficient experimental design, insufficient chart production, and lack of graphics to illustrate the overall structure of the algorithm.

---

> ### Author Response · Authors · 2022-11-12
> **Response to reviewer 9dtX**
>
> Thank you for your review.
>
> Could you clarify in more detail what is meant by
> >*"the author lacks sufficient analysis of model error and policy shift and lacks a detailed description of how the model is established"*?
>
> We discuss the structure of our model in detail in Section 5, Algorithm 1 and Fig 3. We will also make the code publicly available once the paper is accepted.
>
> >*"Lack of a clear graph to reflect the structure and novelty of the algorithm."*
>
> Figure 3 shows the structure of our model. Could you clarify which aspects need more explanation? Or what is unclear?
>
> >"Experimental results do not have multiple sets of seeds to eliminate probabilistic errors."
>
> Note that we have 3 random seeds in the paper and are currently working on adding 6 more seeds for MERMADE. We report the mean and standard error values in Table 1.

---

> ### Author Response · Authors · 2022-11-19
> **More results with 3 additional random seeds**
>
> Hi,
>
> Please note that we added more results with **3 additional random seeds** -- see Table 6 (results in blue) in the Appendix. The results are similar to the original set with 3 random seeds. This shows that our results are stable and consistent with more random seeds.
>
> Thank you for your time!

---

### Official Review · Reviewer_4J8Y · 2022-10-31

**Confidence:** 3
**Correctness:** 3
**Technical Novelty And Significance:** 3
**Empirical Novelty And Significance:** 3
**Recommendation:** 6

**Clarity, Quality, Novelty And Reproducibility:**

The paper is clear and is novel as far as I can tell.
Most of the experiment setup and description of the algorithm is present in the main text, making it easier to reproduce the results.

**Strength And Weaknesses:**

Strengths:
The paper is well-written.
Presents the idea of mechanism design with MERMADE in a clear manner.
Experiment setups and results are well-presented and the performance of the approach is demonstrably strong wrto baselines.

Weaknesses:
Experiments are limited to Matrix games and Bandit settings.
More random seeds and baselines are needed in the experiments.

**Summary Of The Paper:**

The paper explores the problem of mechanism design which studies how to design reward functions and environmental rules defining mathematical games. The applications of mechanism design spans across many domains from optimizing social welfare with economic policies to designing governmental policies. The conventional problem in this space is that it is often expensive to understand the effect of changes to a mechanism design in the real world. Thus, it is often convenient to study the mechanisms in simulations before deploying them in real world.

The paper presents a deep RL approach to mechanism design that learns a world model and uses meta-learning to learn a mechanism that can be adapted quickly to unseen test agents. The approach called MERMADE consists of a planner that has an associated cost for intervening an agent and the goal of this planner is to achieve the designer’s objective. The learning agents maximize the rewards they experience from an environment.

The approach is evaluated on one-shot adaptation performance of the planner in matrix game setups.



**Summary Of The Review:**


MERMADE is interesting because it merges meta-learning with mechanism design. The idea looks very promising.
How do these ideas scale to larger/challenging domains (for example, in MDPs with continuous state spaces, with multi-agents environments)? What kind of research questions need to be addressed to make this idea to scale?

How about including a baseline that is not adapted at test time? What does the performance of the baseline look like when it is evaluated in a zero-shot manner at test time? How large is the gap between MERMADE and this baseline? This will be useful to understand the contribution made by MERMADE over a baseline that is trained similarly but held fixed at test time.

The experiment results seem to be averaged across 3 random seeds. The experiment setup should run fast and should not be a challenge to report results from many more random seeds. Have the authors considered looking into reporting results from more seeds?

How are the hyperparameters tuned for MERMADE and for the baseline methods?

In the experiments, it seems like it is possible to measure the optimal performance of an oracle in mechanism design. It would be interesting to see the difference between MERMADE and such an oracle in the experiments.

---

> ### Author Response · Authors · 2022-11-12
> **Response to reviewer 4J8Y**
>
> We thank you for your time and thoughtful review. We address your primary concerns as follows:
>
>
> *"How do these ideas scale to larger/challenging domains?"*: MERMADE combines a trained world model with a meta-trained policy network for the mechanism designer to decide when and how to intervene. In continuous state space MDPs for the agent, we would still expect our recurrent world model to learn the agent’s transition dynamics from the input actions since the recurrent neural network learns a continuous latent state. However, we might expect the neural network training to require more sample trajectories to learn the dynamics in the continuous state space. Recent work in learning world models using transformers [1] could be a possible alternative approach. For multi-agent environments, the world model has to be trained to learn a representation of the interactions between the agents which might be more challenging than learning the explore-exploit tendencies of a bandit agent.
>
> **Hyperparameters**: We use Adam with lr 0.01 for the policy network in MF-RL and WM-RL. For the policy network in MF-MAML  and MERMADE, the inner optimization loop uses SGD with lr 7e-4 and the outer optimization loop uses Adam with lr 0.001. The world model is trained using Adam with lr 0.01. The learning rates were chosen after a grid search over 0.1 - 0.0001.
>
> **Additional experiments**: We are currently working to report the results for the suggested baseline that is trained similar to MERMADE but held fixed at test time. We will also try to report results for MERMADE with 6 additional random seeds, but based on the mean and standard error values in Table 1, we do not expect a significant variance in performance.
>
> **Comparison to oracle**: Our problem setting in the bandit experiments does not assume access to the agent's base reward or their learning algorithm. But if one were to cheat by having access to these properties for the agent then we could construct a trivial rule based (RB) oracle as described in Sec B.3 and Table 5 in the appendix. Note that such an oracle is not realistic in our mechanism design framework, so we do not compare them to the baselines in Table 1.
>
> [1] Micheli, V., Alonso, E., & Fleuret, F. (2022). Transformers are sample efficient world models. arXiv preprint arXiv:2209.00588.

---

> ### Author Response · Authors · 2022-11-19
> **Additional results with more random seeds**
>
> Hi,
>
> Please note that we added more results with **3 additional random seeds** -- see Table 6 (results in blue) in the Appendix. The results are similar to the original set with 3 random seeds. This shows that our results are stable and consistent with more random seeds.
>
> Thank you for your time!

---

### Author Response · Authors · 2022-11-12
**Comments about our problem setup**

We thank the reviewers for their time and thoughtful comments. We are encouraged that the reviewers appreciate our learning based approach to mechanism design. We would like to provide some more context regarding the general problem of mechanism design and our particular problem setup of adaptive mechanism design in a single agent setting.


Mechanism design studies the problem of inducing desirable outcomes in systems with self-interested agents [1] and in general, it is an NP-hard problem [2]. Prior work in mechanism design has looked at both multi-agent settings (eg. auctions [3], designing different incentive structures in social dilemmas [4]) as well as single-agent settings (eg. revenue maximization [5], matching individuals to jobs in gig economy platforms [6]). Compared to prior approaches, we propose a learning based mechanism design framework in the single-agent mechanism design setting where the agents are themselves adaptive learners. Our problem setup is most similar to the dynamic mechanism design framework in [5] which shows that the optimal mechanism design problem is NP-hard even in a simple deterministic scenario. In this work, MERMADE highlights some of the challenges faced by the mechanism designer in a sequential interaction setup with adaptive learning agents and under incomplete agent information. We empirically show the feasibility of a model-based meta-learning approach to mechanism design in this setup and demonstrate that MERMADE outperforms competing learning based baselines in terms of 1-shot generalization with test agents having different base rewards (or intrinsic preferences) and different adaptive learning characteristics compared to the agents seen during training time.


[1] Hartline, Jason D. "Mechanism design and approximation." Book draft. October 122.1 (2013).

[2] Conitzer, V., & Sandholm, T. (2002). Complexity of mechanism design. arXiv preprint cs/0205075.

[3] Myerson, Roger B. "Optimal auction design." Mathematics of operations research 6.1 (1981): 58-73.

[4] Macy, Michael W., and Andreas Flache. "Learning dynamics in social dilemmas." Proceedings of the National Academy of Sciences 99.suppl_3 (2002): 7229-7236.

[5] Papadimitriou, Christos, et al. "On the complexity of dynamic mechanism design." Games and Economic Behavior (2022).

[6] Pollner, Tristan, et al. "Improved Online Contention Resolution for Matchings and Applications to the Gig Economy." arXiv preprint arXiv:2205.08667 (2022).

---

### Author Response · Authors · 2022-11-19
**Summary of updates during the rebuttal period**

We thank the reviewers for their time and the detailed comments and discussion that helped us improve the paper during the rebuttal period. In summary, we have made the following changes to our submission:

1. Following the comments of reviewer C92e, we have updated the messaging and scoping of the paper to better describe our studied problem setup.
    - We have updated the title of the paper and renamed our proposed approach to MERMAIDE (Meta-learning for Model-based Adaptive Incentive Design).
    - We have mainly updated parts of the text in the abstract and introduction to make the messaging consistent across the paper.
    - We have replaced instances of the term "planner" with "principal" throughout the text.

(Our supplementary material submission (.zip) includes a highlighted comparison of the textual differences in the main paper.)

2. Following the reviewers' suggestions, we have included
    - results for 3 additional seeds for MERMAIDE in the $K=1$-shot evaluation setting,
    - as well results for the $K=0$-shot evaluation setting with **same** and **different types** of bandit agents between training and test time.

    These are included in Table 6 and Table 7 in Appendix C in our updated submission. For the few experiments that have not yet been completed in the rebuttal period, we will update the final values in the camera-ready version if accepted. But based on our preliminary results, we do not expect any significant variation from the currently reported observations.

---

### Decision · Program_Chairs · 2023-01-20

**Decision:**

Reject

**Justification For Why Not Higher Score:**

The paper went through substantial changes, bringing inconsistencies and requiring another round of reviewing.

**Justification For Why Not Lower Score:**

N/A

**Metareview: Summary, Strengths And Weaknesses:**

This paper addresses a general sum game where a principal nudges adaptive agents to align goals with that of the principal by interacting with the agents while having a constraint of minimizing the number of interactions. The proposed method MERMAIDE learns a world model and uses gradient-based meta-learning to learn a principal policy that can be quickly adapted to perform well on unseen out-of-distribution agents.

The paper has an interesting formulation of an important problem. Reviewers pointed out many limitations of the work, such as framing the problem as mechanism design, connection to inverse reinforcement learning, and inclusion of additional baselines, which resulted in substantial changes in the paper. The changes warrant another round of reviewing. Reviewers also showed concern that the latest draft may have significant inconsistencies due to many changes. For example, the first paragraph of the introduction still builds up framing for mechanism design. The work would also benefit from using RL baseline algorithms stronger than Reinforce. I strongly encourage the reviewers to resubmit the paper to the next venue after careful editing.